

**Estimating the open biomass burning emissions in Central and**
**Eastern China from 2003 to 2015 based on satellite observation**
Jian Wu [1], Shaofei Kong [2], Fangqi Wu [2], Yi Cheng [2], Shurui Zheng [2], Qin Yan [1], Huang Zheng [2], Guowei Yang [2],
Mingming Zheng [1], Dantong Liu [3], Delong Zhao [4] and Shihua Qi [1,5]
[1] Department of Environmental Science and Technology, School of Environmental Studies, China University of
Geosciences, Wuhan, 430074, China
[2] Department of Atmospheric Sciences, School of Environmental Studies, China University of Geosciences, Wuhan,
430074, China
[3] Centre for Atmospheric Sciences, School of Earth and Environmental Sciences, University of Manchester,
Manchester M13 9PL, UK
[4] Beijing Weather Modification Office, Beijing 100089, China
[5] State Key Laboratory of Biogeology and Environmental Geology, China University of Geosciences, Wuhan,
430074, China
*Correspondence to:* Shaofei Kong (kongshaofei@cug.edu.cn); Shihua Qi (shihuaqi@cug.edu.cn)
**Abstract.** Open biomass burning (OBB) has significant impacts on air pollution, climate change and potential
human health. OBB has raised wide attention but with few focus on the annual variation of pollutant emission.
Central and Eastern China (CEC) is one of the most polluted regions in China. This study aims to provide a state-of
the-art estimation of the pollutant emissions from OBB in CEC from 2003 to 2015, by adopting the satellite
observation dataset (the burned area product (MCD64Al) and the active fire product (MCD14 ML)), local biomass
data (updated biomass loading data and high-resolution vegetation data) and local emission factors. Monthly
emissions of pollutants were estimated and allocated into a $1 \times 1$ km spatial grid for four types of OBB including
grassland, shrubland, forest and cropland. From 2003 to 2015, the emissions from forest, shrubland and grassland
fire burning had a minor annual variation whereas the emissions from crop straw burning steadily increased. The
cumulative emissions of OC, EC, $CH_4$, $NO_X$, NMVOC, $SO_2$, $NH_3$, CO, $CO_2$ and $PM_{2.5}$ were $3.64 \times 10^3$, $2.87 \times 10^2$,
$3.05 \times 10^3$, $1.82 \times 10^3$, $6.4 \times 10^3$, $2.12 \times 10^2$, $4.67 \times 10^3$, $4.59 \times 10^4$, $9.39 \times 10^5$ and $4.13 \times 10^2$ Gg in these years,
respectively. For cropland, corn straw burning was the largest contributor for all pollutant emissions, by 84%-96%.



Among the forest, shrubland, grassland fire burning, forest fire burning emissions contributed the most and
emissions from grassland fire was negligible due to few grass coverage in this region. High pollutant emissions
were populated in the connection area of Shandong, Henan, Jiangsu and Anhui, with emission intensity higher than
100 ton per pixel, which was related to the frequent agricultural activities in these regions. The monthly emission
peak of pollutants occurred in summer and autumn harvest periods including May, June, September and October, at
which period ~50% of pollutants were emitted for OBB. This study highlights the importance in controlling the
crops straw burning emission. From December to March of the next year, the crop residue burning emissions
decreased, while the emissions from forest, shrubland and grassland exhibited their highest values, leading to
another small peak emissions of pollutants. Obvious regional differences in seasonal variations of OBB were
observed due to different local biomass types and environmental conditions. Rural population, agricultural output,
local burning habits, anthropological activities and management policies are all influence factors for OBB
emissions. The successful adoption of double satellite dataset for long term estimation of pollutants from OBB with
a high spatial resolution can support the assessing of OBB on regional air-quality, especially for harvest periods or
dry seasons. It is also useful to evaluate the effects of annual OBB management policies in different regions.
**Keywords:** emission estimation; open biomass burning; Central and Eastern China; high spatial-temporal resolution;
satellite dataset
**1. Introduction**

Open biomass burning (OBB), which includes forest, shrubland, grassland and crop residues fire burning (van

der Werf et al., 2010; Qiu et al., 2016), is one of the most important sources of gaseous and particulate matter (PM)
especially for fine particulate matter ($PM_{2.5}$) and associated carbonaceous aerosols (elemental carbon (EC) and
organic carbon (OC)) (Zha et al., 2013; Yan et al., 2014; Zong et al., 2016; Zhou et al., 2017). Previous studies have
shown that the OBB contributed to approximately 40% of the average annual submicron EC emission and 65% of
primary OC emission globally (Bond et al., 2013), and impacted more than 45% of $PM_{2.5}$ concentration on days of
heavy air pollution (Deng, 2011). The pollutants of high emission amounts from OBB have significant impacts on
regional and global climate change, air quality and human health (Seiler and Crutzen, 1980; Crutzen and Andreae,
1990; Andreae and Merlet, 2001; Bond et al., 2004; Akagi et al., 2011; Zhang et al., 2016).

From the research in 1970s (Crutzen et al., 1979), multi-scale estimation of biomass burning emissions has

been a research hot topic from global (Seiler and Crutzen et al., 1980; Levine, 1995; Liousse et al., 1995; Bond et



al., 2004; Randerson et al., 2012; Kaiser et al., 2012) to regional scale (Yevich and Logan, 2003; Chang et al., 2010;
Liousse et al., 2010; Li et al., 2017). China is suffering from severe air pollution with hundred millions of open
biomass burned each year (Zhang et al., 2015). The quantitative estimation of pollutants emission for the whole
China (Streets et al., 2003; Tian et al., 2002; Cao et al., 2005; Zhou et al., 2017) or a certain region (Liu et al., 2015;
Zhou et al., 2015; Jin et al., 2017) is also a vital practice, which is the base for assessing the impact of OBB on air
regional quality deterioration. The Central and Eastern China (CEC), including the Central China (Hunan, Henan
and Hubei) and the Eastern China (part of the North Plain of China (Anhui and Shandong), the Yangtze River delta
(YRD, including Zhejiang, Jiangsu and Shanghai) and part of the Pan-Pearl River delta (Fujian and Jiangxi))
(Figure 1), is an area with plenty of vegetation coverage (Figure S1). Yin et al (2017) indicated that the crop residue
fire burning in summer harvest time can lead to the increase of $PM_{2.5}$ concentration in China's middle-east region.
As one of the most heavily polluted regions in China (Chang et al., 2009; Fu et al., 2013), many large cities are
included in this region, such as Nanjing, Wuhan, Shanghai and Hangzhou. Former studies have highlighted the role
of OBB on worsening air quality regionally or at megacities, especially for crop residue burning at harvest periods
(Yamaji et al., 2010; Zhu et al., 2010; Yin et al., 2011; Huang et al., 2012b; Su et al., 2012; Cheng et al., 2014;
Zhou et al., 2016; Zhang et al., 2017).
Previous studies mainly focused on crop residue burning emissions with relatively low spatial and temporal
resolution (Yamaji et al., 2010; Huang et al., 2012b), which may limit its adoption in air quality modeling to give
an accurate result. An accurate estimation of monthly emissions from OBB with a long timescale and high spatial
resolution is still limited. It should be noted that, the OBB activities owned spatial-temporal variation properties
and have changed greatly during the last two decades in China, especially for forestland fire burning (Huang et al.,
2011) and crop residue burning, in view of related policies implementation (as listed in Table S1 and Table S2 of
Supplementary File). As a big agricultural country, the Chinese government has placed a high priority on
environmental pollution prevention caused by OBB. From 1965 to 2015, 51 policy documents for crop straw
management has been formulated and 34 policy documents were introduced after 2008 (Chen et al., 2016). Up to
now, few studies have accurate estimated the biomass burning emissions in a long time period (Fu et al., 2013;
Cheng et al., 2014). The role of the pollution prevention policy on the spatial-temporal variation of pollutants
emitted needs to be better clarified.
In addition, most previous studies used the top-down method (Seiler and Crutzen et al., 1980) to estimate
emission amounts from OBB by national or provincial statistical data and then the total emission amounts
pollutants were re-allocated in grids by population, land cover area, or even equal sharing, which is one of the key



reasons for the high uncertainties of OBB emission inventories (Streets et al., 2003; Klimont and Streets., 2007;
Gadde et al., 2009; He et al., 2013; Zhou et al., 2015; Zhou et al., 2017). Quantitative estimates of biomass burning
were highly improved by the satellite observations of fire burned area or active burning fires (Freitas et al., 2005;
Wooster et al., 2005; Roy et al., 2008; Giglio et al., 2008; Roy et al., 2008; Reid et al., 2009; Sofiev et al., 2009;
Giglio et al., 2010; Liousse et al., 2010; Huang et al., 2012; Li et al., 2016). The improvement of spatio-temporal
distribution evolution was achieved by active fire products (e.g., the AVHRR fire count product (Setzer and Pereira,
1991), MODIS active fire satellite products (Cooke et al., 1996) and VIRS fire count product (Ito et al., 2007)). The
burned area detection was improved by burned area products (e.g., GBA2000 product (Ito and Penner, 2004;
Korontzi, 2005), MODIS burned area dataset (Ito et al., 2007) and Global Fire Emissions Database (GFED)
(Randerson et al., 2012)). However, satellite observation also exhibited weakness in estimating fire burning
emissions (Duncan et al., 2003; He et al., 2015). One is the burned area product, which provides fire burned areas
of the whole month, is limited by the lower pixel resolutions. The size of many small burn scars was below the
detection limit of these products (Eva and Lambin, 1998; Laris, 2005; McCarty et al., 2009; Roy and Boschetti,
2009). Therefore, the contribution of small fires to fire burned areas and corresponding fire burning emissions are
still poorly understood (Randerson et al., 2012). The other is the active fire product, which can provide information
on small fire locations, occurrence time and small fire burned area (Prins and Menzel, 1992; Giglio et al., 2006;
Chuvieco et al., 2008; Roberts et al., 2009; Aragao and Shimabukuro, 2010; Bowman et al., 2011; Lin et al., 2012;
Arino et al., 2012). The uncertainty of fire detection was mainly due to the limitation of satellite overpass periods.
To reduce the uncertainty of emission estimation by satellite products mainly raised by the missing of small burning
areas, the combination of two satellite dataset has recently been proved to be an effective practice (Qiu et al., 2016).

The lack of local biomass data (biomass loading data and vegetation speciation data) and local emission

factors could also introduce uncertainty in emission estimates. At present, the local biomass loading data has not
been updated and still needs to be accurately measured. In addition, local high spatial-resolution vegetation
speciation data was also rarely adopted in OBB estimations. Meanwhile, a lot of researches about OBB have used
the same emission factors without considering the various biomass species and combustion conditions (Andela et
al., 2013; Giglio et al., 2013). All these should be considered and improved in the establishment of OBB emission
inventory.

In this study, the multiple satellite data (MCD14 ML and MCD64Al), local high spatial-resolution of

vegetation speciation data, updated local biomass loading data, local emission factors and survey results were used
to estimate historical OBB emissions from 2003 to 2015 in CEC. High spatial-temporal resolution of emission





120 allocation was achieved. The possible driving factors like rural population, economic level, agricultural production,

121 energy and pollution control policies and anthropogenic activities which may impact the spatial distribution and

122 temporal variation of OBB emissions were explored. They had also been overlooked in previous studies (Song et

123 al., 2009; Chen et al., 2013; Shi et al., 2015). The results here will provide scientific evidence for policy making on

124 controlling OBB emission and modeling its regional impact on air quality, climate and human health. The methods

125 are also helpful for other regions for OBB emission estimation.

126 **2. Methods**

127 **2.1 Estimation of burned areas**

128 OBB emissions in CEC were initially estimated based on the local biomass data (biomass loading data and

129 vegetation speciation data), satellite burned area data (Figure S2 and Figure 2), and emission factors. Fire burning

130 emission amounts are calculated by the following equation (Wiedinmyer et al., 2011; Shi et al, 2015).

$$E_i = \sum_{j=1}^{n} BA_{x,t} \times CE \times BL_x \times EF_{i,j} \tag{1}$$

132 where j stands the different aggregated vegetation types; i stands for different pollutant species; $E_i$ is the

133 emission amounts of different pollutants; $BA_{x,t}$ is the total burned area ($km^2$) of aggregated vegetation class in

134 location x and time t; $CE_x$ is defined as the fraction of OBB; BL is the biomass fuel loading (kg) in different

135 location x ; $EF_{i,j}$ is the emission factor of species i for the j vegetation types.

136 MODIS burned area product (MCD64AL: http://modis-fire.umd.edu/) and the MODIS active fire product

137 (MCD14 ML: https://earthdata.nasa.gov/faq#ed-firms-faq) were combined to obtain accurate open biomass burned

138 area data. MCD64Al has a 500 m spatial resolution and monthly temporal resolution, which can accurately detect

139 the burning area at 500 m pixel. Nevertheless, a much lower pixel resolution burning is difficult to detect by this

140 satellite. The detection of burned area was often affected by weather conditions or cloud cover. Therefore, we use

141 MODIS active fire product MCD14 ML as a supplemental tool to obtain the contribution of small fire burned area.

142 The active fire detection method based on thermal anomalies could detect fires as low as 1/20 of a pixel and could

143 identify much smaller burned areas. However, the active fire product existed as the fire points and could not

144 directly obtain the burned area data. In order to obtain smaller burned areas less than 500 m $\times$ 500 m of a land grid

145 cell, the burned areas of small fire were estimated based on the following method (Randerson et al., 2017).

$$BA_{sf\,(i,t,v)} = FC_{out(i,t,v)} \times \alpha_{(r,s,v)} \times \gamma_{(r,s,v)} \tag{2}$$

147 where $BA_{sf}$ is the small fire burned area in grid cell i, month t, and aggregated vegetation class v; $FC_{out}$ is the

148 total number of MCD14 ML active fires outside of the burned area in each 1 km $\times$ 1 km grid cell; $\alpha$ is the ratio of



BA$_{MCD64A1}$ to the number of active fires with or near the MCD64A1 burned area (km$^2$) and α is equal to the value
of surrounding grid cell if BA$_{MCD64A1}$ is equal to 0;  γ is an additional unit less scalar which indicates the difference
between the active fires in MCD64Al burning area and active fires outside the burning area and  γ is assumed
equal to 1 in this research; r denotes the burning region. s indicates the burning period.
**2.2 Biomass fuel loadings**

For forestland, most previous studies used the forest biomass loading data from Fang et al (1996). The forest

biomass loading data in recent years need to be updated. In this study, for forest land, the biomass loading data
between 2003 and 2008 was collected from Fang et al (1996). From 2008-2015, the update forest loading data was
calculated based on the 8$^{th}$ Chinese National Forest Resource Inventory. The forest biomass density data were
estimated (Table 1) by the following equation:
$$B_i = \frac{T_i}{A_i} \qquad\qquad (3)$$

where i stands for different forest species (broadleaf forest, needleleaf forest and mixed forest); B is the

biomass density; T means the total biomass; A denotes the total area of forest.

The total biomass of different forest species were calculated based on the forest stock volume derived biomass

method. The specific calculated method of different forest biomass were derived from previous studies (Fang et al.,
1996; Tian et al., 2011; Lu et al., 2012; Li et al., 2014; Wang et al., 2014; Wen et al., 2014) (Table 2). Meanwhile,
the forest stock volume data and the total area of forest were collected from the 8$^{th}$ Chinese National Forest
Continuous Inventory. As shown in Table 1, the forest biomass density in recent years has changed a lot in recent
years, which highlighted the updates for improving the emission inventories of OBB.

For grassland and shrubland, local biomass density data were also collected (Pu et al, 2004; Hu et al, 2006) as

listed in Table 1. To determine the accurate provincial amounts of crop residue burning, we gathered the production
of different species of crops from the China Statistical Yearbook (NBSC 2003-2015). The detailed data of
crop-specific residue to production ratio (dry matter) were collected from local statistical data (Table 3) and the
updated data for crop straw burned ratio were derived from survey results (Table 4). Using the updated biomass
data, the accuracy of the estimation of OBB emission is expected to be improved.
**2.3 Combustion efficiency**

In previous studies (Wang et al., 2008; Tian et al., 2011), the combustion efficiency (CE) of OBB is mainly set

as a constant, which may bias the emission estimates. To improve the accuracy, for cropland, the CE was set as 0.68
for legumes and 0.93 for other types (Koopmans and Koppejan, 1997; Wang and Zhang, 2008; Zhang et al., 2011).



For grassland and shrubland, the CE of fires at each grid cell was assumed  as a function of forest cover of
corresponding grid cell (Ito et al, 2004; Wiedinmyer et al, 2006): If areas with tree coverage exceeding 60%, the
CE for woody and herbaceous cover was set as 0.3 and 0.9, respectively; the CE was set as 0 and 0.98 for woody
and herbaceous cover with tree coverage less than 40%; for 40-60% tree cover of fires, the CE was defined as 0.3
for woody fuels and the calculation of herbaceous areas was referred to the following equation:
$$CE_s = e^{-0.13 \times TB} \tag{4}$$
where TB stands for the percent tree cover for fires in each grid cell.
**2.4 Emission Factors**
Emission factors (EFs) of different OBB were summarized in Table 5. EFs for cropland burning were mainly
collected from previous research carried out in CEC (Tang et al, 2014); As the lack of EFs research on some crop
species conducted in CEC and forest, grassland and shrubland conducted in China, EFs were collected from similar
researches (Cao et al, 2008; Tian et al, 2011; GFED4, Version 4; Akagi et al, 2011; He et al, 2015). In addition,
some emission factors measured by our previous research in CEC were included in this study.
**2.5 Spatial and temporal allocation**
In order to estimate high spatial resolution of OBB emission in CEC, a high resolution vegetation map (1:1
000 000) as Figure S1 shown together with the burned area of every opening biomass species were used. All the
data were relocated into a 1 km×1 km grid to identify and estimate spatial variations of OBB emission. The
monthly distribution of OBB emissions were estimated based on the monthly burned area of different vegetation
cover types.
The emissions in t-th grid were calculated using the following equation:
$$E_{t, j} = \frac{BA_{t, j}}{BA_{i, j}} \times E_{i, j} \tag{5}$$
Where $E_{t,j}$ is the emissions of different biomass species j in t-th grid; $BA_{t,j}$ is the burned area in t-th grid cell;
$BA_{i,j}$ is the total burn area of different vegetation types in province i; $E_{i,j}$ is the total emission amounts from OBB in
province i.
**2.6 The influence factors for the OBB emission**
Several detailed statistics data in the NBSC were collected, such as the rural population, the per capita net
income of rural residents, agricultural output and forestry output in each province and each year. They may impact
the OBB emission. Correlation analysis between the OBB emissions and these influencing factors were conducted.





Rural population data in 2003, 2004 and 2010 were lack as the detailed data was not reported in NBSC.
**2.7 Uncertainty analysis**
The Monte Carlo method together with the crystal software was used to evaluate the estimation uncertainty
quantitatively of all the pollutant emissions. Pollutant emissions were estimated from 20, 000 Monte Carlo
simulations with a 95% coincidence interval.
**3. Results and Discussion**
**3.1 Accumulated pollutants emission from OBB in CEC**
Table 6 shows the cumulative OBB emission amounts during 2003-2015 and historical emissions from
different provinces were detailedly listed in Table S3. By the end of 2015, the cumulative emissions of OC, EC,
$CH_4$, $NO_x$, non-methane volatile organic compounds (NMVOCs), $SO_2$, $NH_3$, CO, $CO_2$ and $PM_{2.5}$ were $3.64 \times 10^3$,
$2.87 \times 10^2$, $3.05 \times 10^3$, $1.82 \times 10^3$, $6.4 \times 10^3$, $2.12 \times 10^2$, $4.67 \times 10^3$, $4.59 \times 10^4$, $9.39 \times 10^5$ and $4.13 \times 10^2$ Gg,
respectively. In the following section, for better revealing the spatial-temporal variation of OBB emissions, the
$PM_{2.5}$ variation was detailedly discussed as an example. At the province level from 2013 to 2015, the highest
emission amounts of $PM_{2.5}$ were found in Henan and Shandong, accounting for 28% and 24% of the total emission
amounts, respectively. The lowest emission appeared in Zhejiang and Shanghai, which only contributed for 4% and
0.4%. For other provinces, Hunan, Hubei, Fujian, Anhui, Jiangxi and Jiangsu accounted from 5.5% to 10.1% of the
whole emission.
The contributions of different types of biomass sources for various pollutants were shown in Figure 3a.
Cropland burning contributed the most emission for all the pollutants, from 84%-96%. The forest fire also
exhibited higher emission of $NH_3$, $SO_2$, NMVOC and $PM_{2.5}$, accounting for 12%, 11%,7% and 5% of
corresponding total emission, respectively. As shown in Figure 3b, for the croplands, wheat, corn and rice straw
burning were the top three emission source types for all the pollutants. Corn straw burning contributed the most to
$SO_2$ (48%), $NO_x$ (37%), NMVOCs (33%), CO (32%) and $CO_2$ (28%) emission. Highest contributions of EC (45%),
OC (33%) and $CH_4$ (32%) from rice straw burning was found, while wheat straw burning contributed the most
(31%) to $PM_{2.5}$ emission.
In Figure 4, except for Fujian, cropland burning emission was the largest contributor to the $PM_{2.5}$ emission,
with the contributions ranging from 78% (Hunan) to almost 100% (Shanghai). The higher rural agglomeration,
abundant crops production and more cropland residue burning activities in these provinces can explain the higher
contributions. In Shanghai, one of the most developed cities in China, the highest contribution of cropland burning
is not related with the high levels of agricultural activities, but is only due to the lack of emissions from other open





biomass burning sources. Highest contribution from the forest fire burning and shrubland fire burning were found
in Fujian as 46% and in Hunan as 21%, respectively. For forest fire burning, the provinces located in South China
exhibited higher values, varying from 6% (Hubei) to 44% (Fujian) and for shrubland fire burning, the contributions
varied in 1.5% (Hubei)-7.5% (Zhejiang). While for the Northern provinces (Shandong, Henan, Jiangsu and Hubei),
the contributions ranged around 0.03% and 1%, respectively, which can be neglected. This is mostly due to the
suitable weather conditions, a relative large forest and shrubland coverages and frequent human forestry activities
in those provinces as Figure 2 shown. $PM_{2.5}$ emissions from grassland were negligible with the following provinces
holding the higher contributions: Jiangxi (0.8%), Hunan (0.25%), Anhui (0.1%) and Fujian (0.1%).
From Figure 5, emissions from wheat and corn straw burning mainly concentrated in Shangdong and Henan
(totally accounting for 82% and 78% of the whole emissions, respectively) and the rice straw burning exhibited
higher concentrations in Hunan, Jiangxi and Hubei provinces, by 25%, 18% and 16%, respectively. The total
contributions of rapeseed, cotton, potato and peanut straw burning to the $PM_{2.5}$ emission were relatively small,
occupied by 21%-24% of the total emissions. Most emissions from cotton, peanut and potato straw burning located
in Shandong (totally accounting for 35%, 35% and 20%) and Henan (totally accounting for 19%, 40% and 15%),
Hubei (32%) and Hunan (31%) were the major provinces for rapeseed straw burning emissions. In addition,
emissions from soya bean, sugar cane, tobacco, sesame and sugar beet straw burning were negligible, which never
exceeded 1% of total crop emission in this study.
**3.2 Temporal variation and spatial distribution for OBB emissions in CEC**
**3.2.1 Yearly variation**
Historical emissions of OBB from 2003 to 2015 in CEC were shown in Figure 6. The multi-year variation
tendency of OBB emissions for various pollutants was similar. Take $PM_{2.5}$ as example, emissions exhibited clearly
increasing trends from 2003 (256 Gg) to 2008 (353 Gg) and then decreased in the following two years to 322 Gg.
After 2010, there existed higher (2011, 2013 and 2015) and lower values (2010, 2012 and 2014) alternately. The
values in 2011, 2013 and 2015 all did not exceed the peak values in 2008.
In 2008, intensive policies for utilization of straw energy (Table S1) and strengthening the forest fire and
grassland fire prevention (Table S2) were published, which effectively limited the emissions from forest and
shrubland burning as Figure 7a shown. Peak emissions for $PM_{2.5}$ from forest, shrubland and grassland burning were
found in 2008, as 49 Gg, 8.9 Gg and 0.7 Gg, respectively. Obvious decreasing was found from 2008 to 2010, to 19
Gg, 4.8 Gg and 0.24 Gg, respectively. Then they exhibited inter-annual oscillation from 2010 to 2015, with higher
emission amounts in odd years and lower emission amounts in even years (Jin et al., 2017a). The multi-year





tendency for forest, shrubland and grassland were mainly affected by the variations in climate, management
measures and other human forcing. It can also conclude that the yearly variation trends of pollutants from OBB
were mainly impacted by the emission from forest, shrubland and grassland burning, but not the crop burning.
The emissions of $PM_{2.5}$ from crop burning exhibited quite different yearly variation trend with other three
types of biomass burning, which gradually increased from 2003 (228 Gg) to 2015 (323 Gg), by 29%. The increase
of crop residue production can explain the increasing of pollutant emission. Meanwhile, from Table S1, the
controlling of pollutants from crop residue burning in China started from 1970s, and in 2000, the law for prevention
of air pollution was published. Then in 2003, the regulations on straw banning and comprehensive utilization were
released. From 2005 to 2015, all the policies were to improve the straw energy utilization to reduce the air pollution
raised by its burning. However, it has to say, the policies may not be well implemented, with the annual averaged
increasing amounts of 7.3 Gg for $PM_{2.5}$. From Figure 7b, the large contributions to $PM_{2.5}$ (22%-28% and 29%-33%)
and increasing trends for corn burning and wheat burning could be found, which should be further focused. The
contributions from rice burning slightly decreased in the past decade, by about 19% from 2003 to 2015. Other types
of biomass totally accounted for averaged 25% of $PM_{2.5}$ emission and all exhibited slightly decreasing trend from
2003 to 2015, increased by about 21%-29%.
Figure 8 showed that the crop burning emission in Henan, Shandong, Anhui, Jiangsu, Hubei, Hunan and
Jiangxi exhibited obvious increasing trends, which suggested the importance of crop burning control in these
provinces. For Fujian and Zhejiang, no obvious increase for crop burning emissions was found, implying that the
emissions have been well controlled in these years. It should be noted that in Fujian and Zhejiang, the main crop is
rice while in other provinces, the main crops are corn and wheat especially for Northern provinces. To conclude,
pollutants emitted from crop residue burning (wheat, corn and rice) are still now the key sources for air pollution, in
view of its increasing emission trend. The randomness of burning activities and corresponding widespread and
scattering distribution make it difficult to control them. The wheat and corn emissions at Northern provinces and
rice burning emissions at Southern provinces should be controlled specially in the future.
In Figure 10, the $PM_{2.5}$ emission from crop residue burning exhibited higher amounts for Henan and Shandong
province in 2015, as 100 Gg and 82 Gg, respectively, which are 200% -1200% times of those for other provinces.
As the main source regions for air pollution of Yangtze River Delta (YRD) and Beijing-Tijin-Hebei (BTH) region
(Fu et al., 2013; Zhou et al., 2015), the enforced and effective control of crop residue burning in the two provinces
at summer and autumn harvest periods are important for improving the air quality of these regions.
**3.2.2 Monthly distribution**





The monthly $PM_{2.5}$ emission variation of different OBB in CEC was shown in Figure 9a. The total monthly
$PM_{2.5}$ emission held higher amounts in May and June (90.4 Gg-179.3 Gg), followed by December to March of next
year (32.2 Gg-127.3 Gg) and September-October (8.2 Gg-89.2 Gg), and was lowest during July-August (14.3
Gg-65.9 Gg). As the emission amounts of crop fire burning were one or two magnitude higher than other three
types of biomass burning, the month variation of total $PM_{2.5}$ variation was mainly controlled by the crop biomass
burning (Zhang et al., 2016). The periods with highest $PM_{2.5}$ emissions were just the summer and autumn harvest
times, when the burning activities are more frequent. The peak of open biomass fire burning occurs in May and
June totally accounted for 42% of the whole $PM_{2.5}$ emission in 2003-2015, which is caused by the harvest of winter
wheat, especially in Henan, Shandong, Jiangsu and Anhui (Figure 9b). Large amounts of wheat straw were burned
after the harvest to increase the soil fertility and prepare for following corn cultivation (Levine et al., 1995).
Though the open biomass burning was strictly forbidden in recent years, scattered burning activities still existed in
China. The small peak of open biomass burning emission in September to October (totally accounted for 13.8% of
the whole $PM_{2.5}$ emission in 2003-2015) can be attributed to the burning of corn straw after corn harvest. As shown
in Figure S4, in recent years, the emissions in CEC and major agricultural province during harvest time have shown
a rapid decline, in accordance with the change tendency of burned area due to increased government management.
Considering of the increase tendency of crops straw burning from year to year, it is worth noting that fire burning
out of harvest season as a way of circumventing governmental polices needs to continue to be well regulated. From
December to February of the next year, the crop residue burning emissions decreased to the lowest level in the
whole year (18.9% of the whole $PM_{2.5}$ emission in 2003-2015). However, during December to March, the
emissions of $PM_{2.5}$ from forest, shrubland and grassland exhibited their peak values, totally occupied by 67% of the
whole $PM_{2.5}$ emission of forest, shrubland and grassland fire burning in 2003-2015.
Figure 10 clearly listed the monthly average emissions of $PM_{2.5}$ from OBB in different provinces. These
provinces were classified based on the correlation of emissions in each month of 2003-2015. Henan, Shandong,
Anhui and Jiangsu provinces ($R^2$ higher than 0.92, P<0.01), as one of the largest and contiguous wheat planting
areas in China (Fang et al., 2014), have two crop rotations. The highest monthly emissions were observed for
winter wheat harvesting (sown in October and harvested from May to June) and corn harvesting (sown in middle
June and harvested from September to October). A large proportion of crop straw is always burnt directly after the
crop harvest (MEPC, 2015). For Hubei province, agricultural emissions fluctuated over the period from February to
October with several peaks due to that different crop species matured in succession. In Jiangxi, Fujian and Hunan
($R^2$ higher than 0.9, P<0.01), the largest monthly emissions were observed with forest and shrubland fire burning





during the time between December and March in the next year, which is the dry seasons in these provinces (Li et al.,
2014; Li et al., 2015). And in other months, the emissions were limited. For Shanghai and Zhejiang ($R^2 = 0.7$,
P<0.01), lowest levels of $PM_{2.5}$ emissions were found, with peak values also occurred in summer and autumn
harvest periods. Obvious two peaks were found for April-May and July-August periods, which may reflect the rice
harvesting at these times. To sum up, these regional differences of monthly $PM_{2.5}$ emissions from OBB were
mainly caused by the different biomass burning types and times as well as corresponding environmental conditions.
**3.2.3 Spatial distribution within 1 km×1 km of $PM_{2.5}$ emitted from OBB in CEC**
The spatial distribution of $PM_{2.5}$ emitted from OBB within 1 km×1 km resolution was mapped based on the
burned area and a high-resolution vegetation map (1:1 000000) in CEC. The multi-year averaged spatial
distributions of $PM_{2.5}$ emission are shown in Figure 11. It can be found that the OBB was widespread and scattered.
The average emissions intensity of $PM_{2.5}$ ranged from 0 to 15 tons per pixel in most provinces. The variation range
is mainly caused by the social-economic development level, rural population and agricultural activities. The highest
value in different provinces was all caused by the crops fire burning due to the centralized burning of them in a
relatively small area. Some pixels with high emissions exceeding more than 100 tons each year were found in
Henan, Shandong and Hunan. It can be attributed to the large amounts of crop straws in these provinces. The pixels
of high emission intensity more than 70 tons from crop straw burning were also found in Hubei, Jiangsu and Anhui.
For forest and shrubland fire burning, the high emission points from (more than 30 tons per pixel) were found in
Fujian and Jiangxi. Lower emission intensities in Zhejiang (lower than 10 tons per pixel on average) and Shanghai
(lower than 7 tons on average) were mainly due to the highly developed economy and limited agricultural activities
(Su et al., 2012). In addition, northern Anhui and eastern Jiangsu were found high emissions of OBB with a
relatively lower intensity (lower than 15 tons per pixel on average), which may be due to that the crop straw was
burned in a large area in these regions.
Though the emission intensities varied in the past ten years, the areas with high emission amounts are
uniformed. They were mainly located in the main agricultural areas in eastern Henan, southern Shandong, northern
Anhui, northern Jiangsu, eastern Hubei and northern Hunan. This result is in accordance with formers (Huang, et
al., 2012b). The junction regions of the four provinces-Henan, Shandong, Anhui and Jiangsu should be paid more
attention, where the pollutants emission from OBB jointed together. This was similar to a recent research (Jin et al.,
2017b). This region belongs to HuangHuai Plain, with large area of croplands and low economic development
levels. The opening burning activities and corresponding banning policies are both abundant in village scale. The
game of "cat and mouse" is frequently acted. More effective policies for guiding or helping farmers to utilize straw



energy rather than banning crop residue burning arbitrarily should be considered sincerely. In Zhejiang and
Shanghai, OBB emissions are sparsely scattered, due to the relatively developed economic level, scarce biomass
sources and low agricultural activities. The recycling of crop straw faces many difficulties due in part to its high
cost and the relative low price of crop straw. Improving policies for effectively utilizing crop residue straw is also
an important challenge for the government.
Figure 12 highlights the spatial distribution of PM$_{2.5}$ emitted from OBB in different seasons of 2015.
Emissions were more concentrated in summer, followed by winter. In summer, the emission was mainly
concentrated in the connection regions of Henan, Shandong, Anhui and Jiangsu, mainly raised by crop straw
burning as discussed before. In winter, Jiangxi, Hunan and Fujian showed the higher emission intensities from
forest and shrubland burning.
**3.3 The impact of social-economic factors on OBB emission**
Emissions from OBB were found to be in line with the local burning habit, anthropogenic activities, rural
population, local economic level, agricultural level and pollution controlling policies. Local burning habits have a
great influence on different types of OBB emissions. In our survey, in agricultural provinces, such as Henan,
Shandong, Jiangsu and Anhui, people always burn crop straws in sowing and harvest seasons. Despite the strict
implementation of crop residue burning management policies, the burning habit is difficult to change in a short
time. Less crop residue production and crop burning activities are found in Jiangxi and Fujian, where people are
accustomed to use crop straw to feed draught animals and produce biogas instead of open burning directly. The
emission from crop residue burning is low. However, due to the rich forest and shrubland resources, wood is served
as the staple household fuel, which mainly comes from felling trees or collecting branches. These human activities
can lead to an increase in forest and shrubland burning, resulting in the elevated levels of corresponding emission in
these provinces.
Some anthropogenic activities also pose impact on OBB emissions. Biomass burning emissions in April can
be enhanced by human burning activities in the tomb-sweeping day. The tomb-sweeping day (often in April 4 or
April 5) is a time to memorize the death. People sweep their graves and burn sacrifices by ignited straw, which can
easily cause grass, shrub and forest fires (Qiu et al., 2016). The fire points at the tomb-sweeping day can occupied
by 22%-38% of the whole file points in April in CEC in some years (Figure S3). The Chinese government has also
introduced policies to prevent forest, shrubland and grassland fires on tomb-sweeping day (Table S2). The wildfires
caused by biomass burning from late January to early February are partially related to the firework burning in the
Spring Festival (Zuo, 2004). The firework burning activities for celebration and official sacrifices to ancestors in





the Spring Festival can easily lead to grass, shrub and forest fires. All these activities can affect the emission levels
in a short time scale.
In order to understand the impact of the rural population, local economic level and agricultural level,
correlation analysis between $PM_{2.5}$ emissions from OBB and statistics data (the rural population, the per capita net
income of rural residents and agricultural output (crop straw burning) and forestry output (forest, shrubland and
grassland burning)) in different provinces were conducted. For crop residue burning, significant positive
correlations were found between the rural population, agricultural output and the $PM_{2.5}$ emissions from crops straw
burning for the whole CEC (Figure 13a). The high rural population and agricultural output indicates that
agricultural activities are quite important in a certain region. With more crops residue produced, it can easily cause
high emissions from cropland fire burning. No significant correlations were found for $PM_{2.5}$ emission from crop
straw burning with the income of rural residents (Figure 15), which indicates that the rural economic level in
different regions in CEC have no relationship with the $PM_{2.5}$ emission. Then we calculated the correlations between
the change tendency of $PM_{2.5}$ emission from crops fire burning and the multi-year variation of other three
social-economic factors as Table 7 shown for different provinces. Significant positive correlations were found for
$PM_{2.5}$ emission and per capita income of rural residents and agricultural output (most $R^2$ higher than 0.58, P<0.01)
and negative correlation were found for $PM_{2.5}$ emission with rural population (most $R^2$ higher than 0.7, P<0.01)
except for the provinces of Shanghai, Zhejiang and Fujian, which are underdeveloped agricultural provinces. From
2003 to 2015, with the increase of agricultural outputs, more crop residue was produced. However, rapid economic
development and less rural population in each province lead to the popular of commercial energy and clean energy
in rural area. It decreased the demands in using crop residue as fuel. As a consequence of this, more crop residues
were directly burned in the agricultural field. But it was not suitable to Shanghai, Zhejiang and Fujian, where holds
less crop residue production and high utilization efficiency of crop straws.
Positive correlations were also found between forestry output and $PM_{2.5}$ emission from forestland, shrubland
and grassland burning (most $R^2$ higher than 0.5, P<0.05) in the whole CEC (Figure 13b), and it indicated that
human forestry activities played positive role on open fire burning (Yan et al., 2006). According to our survey,
human forest activities such as felling trees or picking up branches from trees can easily cause more forest and
shrubland burning. However, compared with the crops straw burning, no correlation was found between $PM_{2.5}$
emission and other statistics data (the rural population and the per capita net income of rural residents) (Figure 13b
and Table S5). It may indicate that the forestry fire burning activities were not predominantly affected by the rural
human living activity. According to previous studies, forestry fire burning was affected by environmental




conditions and human activities with environmental factors have a larger impact (Chen et al., 2013).
**3.4 Comparison with others**
Emission data from OBB in CEC during the past several years have been compared with other studies for the
similar year (Table 8). Compared with the emissions derived from Wang et al. (2008) based on statistical data, the
OC, EC, $CH_4$, $NO_x$, NMVOC, $NH_3$, $CO_2$ and CO emissions are close, with the differences ranging from -41% to
12 %. The difference in $SO_2$ and $PM_{2.5}$ emission is relative high. The differences were mainly caused by the
accuracy of biomass data, the burned ratio for various crop types and the selection of EFs. The results in this study
can decrease the uncertainty from statistical data for forest, shrubland and grassland fire burning, as there are
limited forestry statistical data. Compared with Huang et al. (2012), who use the same emission factors for different
crops straw, the estimate in our study is more accuracy. An obvious underestimation of $PM_{2.5}$ emission from crop
straw burning were found in Jin et al. (2017), in which all the crop species in the study were not considered.
The estimation based on satellite observation was prevalent recently. Compared to Zhou et al. (2017) who
estimated the pollutant emission amounts from MODIS burned area products, the results in this study were much
higher. The reason may be that when using a single satellite data set, pollutant emission can be underestimated due
to some actual agricultural fire activities could not be detected (van der Werf et al., 2010). The lower emission of
$CO_2$, NMVOC, $SO_2$ and NOx in our study is due to the smaller EFs values used. Our emission estimation of the
pollutants is more similar to the results in Qiu et al. (2016), who also used multiple satellite products (MCD14 ML
and MCD64Al) to estimate the OBB emissions of China in 2013, with the differences of the two studies ranging
from -42% to 22%. For $CH_4$, $NO_x$, NMVOC, $NH_3$ and $CO_2$, the differences were less than 10%. The reason for the
differences is due to the use of updated local biomass data and EFs in this study. Meanwhile, the updated forest
loading data also reduced the uncertainty of pollutant emissions from forest fire burning. At the same time, the EFs
used for various biomass burning types, the crop specific residue to production ratio data and the burned ratio for
various crop types were all localized in CEC in this study. The combination of multiple satellite products with local
EFs data and updated local biomass data can improve the estimation of pollutant emission from OBB effectively.
**3.5 Uncertainty analysis**
Emission uncertainties in our study were associated with the fire satellite products, biomass fuel loading data,
combustion efficiency and emission factors. The estimation for large fires was proved to be reliable for burned area
product MCD64AL (Giglio et al., 2013). For the active fire product MCD14ML, the uncertainty was mainly caused
by the satellite passing time. The small fires which burned in 10:30 am-1:30 pm could not be captured by
MCD14ML. The uncertainty of biomass fuel loading data was estimated to be approximately 50% (Shi et al., 2015)





and the uncertainty of EFs of each pollutant ranging from 0.03 to 0.85. At last, in order to evaluate the estimation
uncertainty quantitatively, the Monte Carlo method was used. Pollutant emissions were estimated from 20, 000
Monte Carlo simulations with a 95% coincidence interval. Table 9 shows the emission uncertainty for different
pollutants for each year of 2003-2015. On average, the uncertainty of the estimated OC, EC, $CH_4$, $NO_x$, NMVOCs,
CO, $SO_2$, $NH_3$, $CO_2$ and $PM_{2.5}$ were (-30%, 30%), (-48%, 48%), (-20%, 20%), (-20%, 20%), (-45%, 45%), (-18%,
18%), (-45%, 45%), (-35%, 35%), (-3%, 3%) and (-36%, 36%), respectively.
Compared with previous studies, for the emission estimation of forest burning, the uncertainty was improved
by the updated forest fuel loading data. For cropland, the uncertainty was improved by the adoption of local
grain-straw ratio data and the crop residue burned ratio data based on survey results. Compared with the constant
combustion efficiency in previous researches, the activity combustion efficiency data could also reduce the
uncertainty as they could more accurately reflect the actual combustion conditions (Chen et al., 2013). Meanwhile,
the local measured EFs data for different biomass burning species from previous researches also improved the
accuracy of the estimation. Therefore, due to the adoption of multiple satellite products, local high resolution
vegetation data, updated local biomass distribution data and local emission factors, the estimation of emissions in
our study is relatively more reliable.
**4 Conclusions**
In this study, a combination of the burned area product (MCD64Al) with the active fire product (MCD14 ML),
as well as local high resolution vegetation speciation data, updated local biomass data, local emission factors and
survey results were used to estimate the pollutant emissions from open burning in Central and Eastern China (CEC)
from 2003 to 2015. The emission from crop residue, forest, shrubland and grassland burning were considered.
Crop residue burning was the major source type for pollutant emissions, followed by forest fire and shrubland
burning. The grassland fire burning emissions were negligible in CEC. For cropland, the fire burning was mainly
concentrated in agricultural provinces, such as Henan and Shandong. For forest and shrubland, the fire burning was
mainly concentrated in Fujian, Jiangxi and Hunan provinces, with abundant forest resources. Wheat, corn and rice
straw were the major three types of crop straws. Wheat and corn straw burning dominated in Shangdong and Henan
and the rice straw burning dominated in Hunan, Jiangxi and Hubei provinces. For various pollutant emissions, corn
straw burning was the largest contributor to $SO_2$, $NO_X$, CO, NMVOC, $CO_2$, $NH_3$; OC, EC and $CH_4$ emissions were
mainly produced by rice straw burning; wheat straw burning was the largest contributor to $PM_{2.5}$. The spatial
distribution of opening biomass residue burning in different years is similar. The high emissions are mainly found
in the major agricultural areas in eastern Henan, southern Shandong, northern Anhui, northern Jiangsu, eastern





Hubei and northern Hunan, due to their abundant agricultural activities cultivated areas and low straw utilization
efficiency.

From 2003 to 2015, the multi-year tendency of opening biomass residue burning emission for various

pollutants is similar. Emissions from crop straw burning continued to increase, due to the gradual increase of crop
residue production. While emissions from forest, shrubland and grassland fire burning exhibited minor fluctuation
from year to year, influenced by the environmental conditions, management measures and other human driving
factors. Monthly distributions revealed that the pollutant emissions were at the highest levels in May and June, with
the lowest emissions in July and August. The high emissions from May to June and October were mainly caused by
crop straw burning in sowing and harvest times. It is worth noting that the fire burning activities at harvest season
need to be regulated continuously by local governments and the fire burning out of harvest season should also be
paid more attention in recent years. Meanwhile, emissions from forest and shrubland accounted for the vast
majority of total emissions in December to March of the next year. The rural population, agricultural output and
economic levels impacted on the emission of crop residue burning while the emissions from forestland, shrubland
and grassland burning were more affected by environmental conditions.

The estimation of historical emissions by satellite data in this study will provide a fundamental role in

assessing the role of pollution prevention policies on open burning activities published in the last decade. The
high-spatial ($1 \times 1$ km) resolution monthly emission inventory is also useful in modeling regional air quality and
human health risks in the future.

**Acknowledgements**

This study was financially supported by the Key Program of Ministry of Science and Technology of the

People's Republic of China (2016YFA0602002; 2017YFC0212602), the Key Program for Technical Innovation of
Hubei Province (2017ACA089). The research was also supported by the Fundamental Research Funds for the
Central Universities, China University of Geosciences, Wuhan.

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



**Table 1.** Forest, shrubland and grassland biomass fuel loading (kt km$^{-2}$) in each province.

| Province | Forest (2003-2008)[a] | Forest (2009-2015)[*] | Shrubland[b] | Grassland[c] |
|---|---|---|---|---|
| Shandong | 4.26 | 2.95 | 6.94 | 0.78 |
| Henan | 5.66 | 4.16 | 6.94 | 0.77 |
| Anhui | 6.32 | 3.61 | 12.2 | 0.77 |
| Jiangsu | 4.7 | 2.64 | 6.86 | 0.72 |
| Hubei | 5.34 | 3.28 | 7.87 | 0.88 |
| Hunan | 4.79 | 2.52 | 17.4 | 0.8 |
| Jiangxi | 4.75 | 3.08 | 18.5 | 0.76 |
| Fujian | 6.29 | 5.91 | 18.9 | 0.85 |
| Zhejiang | 3.51 | 3.11 | 18.4 | 0.86 |
| Shanghai | 6.09 | 2.99 | 6.86 | 0.93 |

References: [a] Fang et al. (1996); [b] Pu et al. (2004); [c] Hu et al. (2006); [*] This study.



**Table 2.** Parameters of biomass-stem volume regression functions of dominant tree species in forests (B=aV+b)

| Tree species | a | b | Tree species | a | b |
|---|---|---|---|---|---|
| Larix | 0.967[a] | 5.7598[a] | Cinnamomum camphora | 1.0357[a] | 8.0591[a] |
| Pinus koraiensis | 0.5185[a] | 18.22[a] | Phoebe | 1.0357[a] | 8.0591[a] |
| Pinus sylvestris var. mongolica | 1.11[a] | | Elm | 0.7564[f] | 8.3013[f] |
| Pinus densiflora | 1.0945[b] | 2.004[b] | Robinia | 0.7564[a] | 8.3103[a] |
| Pinus thunbergii parl | 0.5168[b] | 33.237[b] | Schima superba | 0.76[e] | 8.31[e] |
| Chinese pine | 0.7554[a] | 5.0928[a] | Sweetgum | 0.76[e] | 8.31[e] |
| Pinus armandi | 0.5856[a] | 18.7435[a] | Other hard broad leaf | 0.7564[b] | 8.3103[b] |
| Pinus massoniana | 0.52[a] | | Tilia | 0.7975[b] | 0.4204[b] |
| Pinus yunnanensis | 0.52[a] | | Sassafras | 1.0357[a] | 8.0591[a] |
| Pinus kesiya var. langbiamensis | 0.510[b] | 1.045[b] | Populus | 0.4754[a] | 30.603[a] |
| Pinus densata | 0.5168[b] | 33.237[b] | Salix | 0.4754[c] | 30.6034[c] |
| Foreign pine | 0.5168 | 33.2378 | Paulownia | 0.8956[d] | 0.0048[d] |
| Pinus elliottii | 0.51[e] | 1.05[e] | Eucalyptus | 0.7893[a] | 6.9306[a] |
| Pinus taeda | 0.5168[f] | 33.2378[f] | Rich acacia | 0.4754[a] | 30.60[a] |
| Mount huangshan pine | 0.5168[f] | 33.2378[f] | Casuarina equisetifolia | 0.7441[b] | 3.2377[b] |
| Joe pine | 0.5168[f] | 33.237[f] | Melia azedarach | 0.4754[b] | 30.603[b] |
| Other pine | 0.5168[a] | 33.2378[a] | Other soft broad leaf | 0.4754[b] | 30.603[b] |
| Cunninghamia lanceolata | 0.399[a] | 22.54[a] | Coniferous mixed | 0.5168[f] | 33.2378[f] |
| Cryptomeria fortunei | 0.4158[a] | 41.3318[a] | Broad-leaved mixed | 0.8392[b] | 9.4157[b] |
| Metasequoia | 0.4158[a] | 41.3318[a] | Coniferous and broad-leaved mixed | 0.7143[b] | 16.9154[b] |
| Taxodium ascendens | 0.399[a] | 22.541[a] | Betula | 0.9644[a] | 0.8485[a] |
| Abies | 0.4642[a] | 47.499 | White birch | 0.9644[a] | 0.8485[a] |
| Picea | 0.4642[a] | 47.499[a] | Betula costata | 0.9644[a] | 0.8485[a] |
| Tsuga | 0.4158[a] | 41.3318[a] | Water, beard and yellow | 0.7975[b] | 0.4202[b] |
| Keteleeria | 0.4158 | 41.3318 | Manchurian Ash | 0.798[c] | 0.42[c] |
| Cupressus | 0.6129[a] | 26.1451[a] | Juglans mandshurica | 0.798[c] | 0.42[c] |
| Yew | 0.4642[b] | 47.499[b] | Amur corktree | 0.798[c] | 0.42[c] |
| Other fir | 0.399[a] | 22.541[a] | Quercus | 1.3288[a] | -3.8999[a] |

References: [a] Fang et al. (1996); [b] Wen et al. (2014); [c] Lu et al. (2012); [d] Tian et al (2011); [e] Wang et al (2014); [f] Li et al. (2014).

B = aV + b: B indicates the total biomass of different tree species (t); V indicates the forest stock volume ($m^{-3}$).





**Table 3.** The detailed crop specific residue to production ratio data for each province

| Province | Rice | Corn | Wheat | Cotton | Rapeseed | Soy bean | Sugar cane | Peanut | Potato | Sesame | Sugar beet | Tobacco |
|---|---|---|---|---|---|---|---|---|---|---|---|---|
| Anhui | 1.09[a] | 1[a] | 1.12[a] | 3.35[a] | 2.98[a] | 1.52[a] | 0.34[a] | 1.26[a] | 0.53[a] | 2.01[a] | 0.37[a] | 0.71[a] |
| Fujian | 0.85[b] | 1.04[c] | 1.17[c] | 2.91[d] | 2.87[d] | 1.5[d] | 0.43[d] | 1.08[m] | 0.57[d] | 2.01[d] | 0.43[d] | 0.56[d] |
| Henan | 1[c] | 0.96[c] | 1.08[h] | 2.41[i] | 2.87[d] | 1.5[d] | 0.34[d] | 0.89[d] | 0.57[d] | 1.78[d] | 0.43[d] | 0.49[d] |
| Hubei | 1.17[e] | 1.04[c] | 1.17[c] | 4.09[j] | 3.17[k] | 1.5[d] | 0.43[d] | 1.14[d] | 0.57[d] | 2.01[d] | 0.43[d] | 0.71[d] |
| Hunan | 0.94[f] | 1.11[g] | 1.17[c] | 2.91[d] | 3[l] | 1.5[d] | 0.43[d] | 1.38[n] | 0.57[d] | 2.23[d] | 0.43[d] | 0.85[d] |
| Jiangsu | 1.04[a] | 1[a] | 1.41[c] | 2.61[i] | 2.98[a] | 1.52[a] | 0.34[a] | 1.26[a] | 0.53[a] | 2.01[a] | 0.37[a] | 0.71[a] |
| Jiangxi | 1[c] | 1.04[c] | 1.17[c] | 2.91[d] | 2.87[d] | 1.5[d] | 0.43[d] | 1.14[d] | 0.57[d] | 2.01[d] | 0.43[d] | 0.71[d] |
| Shandong | 1[c] | 0.96[c] | 1.33[c] | 2.91[d] | 2.87[d] | 1.5[d] | 0.43[d] | 0.85[d] | 0.57[d] | 2.01[d] | 0.43[d] | 0.71[d] |
| Shanghai | 1.28[a] | 0.93[a] | 1.09[a] | 3.35[a] | 2.98[a] | 1.52[a] | 0.34[a] | 1.26[a] | 0.53[a] | 2.01[a] | 0.37[a] | 0.71[a] |
| Zhejiang | 1.07[a] | 0.96[a] | 1.2[a] | 3.35[a] | 2.98[a] | 1.52[a] | 0.34[a] | 1.26[a] | 0.53[a] | 2.01[a] | 0.37[a] | 0.71[a] |

References: [a] Zhu et al. (2017); [b] Chen et al. (2008); [c] Xie et al. (2011a); [d] Xie et al (2011b); [e] Zeng et al (2007); [f] Ao et al. (2007); [g] Lei et al. (2009); [h] Zhao et al. (2008); [i] Xue et al. (2006); [j] Yu et al (2009); [k] Zou et al (2008); [l] Liu et al. (2010); [m] Tang et al. (2009); [n] Li et al. (2008).

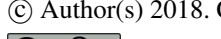



**Table 4.** The detailed crops straw burned ratio data from survey results.

| Region | Crops straw burning percentage |
|--------|-------------------------------|
| Anhui | 0.10[a] |
| Fujian | 0.188[b] |
| Henan | 0.208[c] |
| Hubei | 0.207[c] |
| Hunan | 0.278[c] |
| Jiangsu | 0.10[a] |
| Jiangxi | 0.18[c] |
| Shandong | 0.178[c] |
| Shanghai | 0.148[d] |
| Zhejiang | 0.319[c] |

References: [a] Tian (2011); [b] Huang (2014); [c] Peng et al; (2016). [d] Zhou et al (2017).



**Table 5.** The emission factors of open biomass burning emissions for various pollutants (g kg$^{-1}$ dry matter)

| Vegetation | OC | EC | CO | CH$_4$ | NO$_x$ | NMVOCs | SO$_2$ | NH$_3$ | CO$_2$ | PM$_{2.5}$ |
|---|---|---|---|---|---|---|---|---|---|---|
| Corn | 1.457[*] | 0.14[*] | 70. 2[a] | 4. 4[b] | 3.36[a] | 10[c] | 0.45[h] | 0. 68[g] | 1261[f] | 5[c] |
| Rice | 1. 96[a] | 0. 52[c] | 52. 32[c] | 3. 9[b] | 1.42[d] | 6. 05[f] | 0. 147[a] | 0. 53[g] | 791[f] | 3.03[d] |
| Wheat | 2.7[b] | 0. 49[a] | 61. 90[c] | 3. 4[b] | 1.19[d] | 7. 5[c] | 0. 147[c] | 0. 37[b] | 1557[f] | 7.6[a] |
| Cotton | 3.06[c] | 0. 57[f] | 70.29[c] | 4.4[b] | 2. 98[c] | 10[c] | 0.23[c] | 0. 68[b] | 1445[h] | 11.7[c] |
| Rapeseed | 1. 08[d] | 0. 23[d] | 34.3[d] | 3. 9[b] | 1. 12[d] | 8. 64[c] | 0. 25[c] | 0. 53[g] | 1445[h] | 5.76[c] |
| Soya bean | 1. 05[d] | 0. 13[d] | 32.3[d] | 3. 9[b] | 1. 08[d] | 8. 64[c] | 0. 25[c] | 0. 53[g] | 1445[h] | 3.32[d] |
| Sugar cane | 2.03[c] | 0. 41[c] | 40. 08[f] | 3. 9[b] | 2. 03[c] | 11. 02[f] | 0. 25[c] | 0. 53[g] | 1445[h] | 4.12[f] |
| Peanut | 2.03[c] | 0. 41[c] | 55. 13[c] | 3. 9[b] | 2. 11[c] | 8. 64[c] | 0. 25[c] | 0. 53[g] | 1445[h] | 5.76[c] |
| Potato | 2. 03[c] | 0. 41[c] | 55. 13[c] | 3. 9[b] | 2. 11[c] | 8. 64[c] | 0. 25[c] | 0. 53[g] | 1445[h] | 5.76[c] |
| Tobacco | 2.03[c] | 0. 41[c] | 55. 13[c] | 3. 9[b] | 2. 11[c] | 8. 64[c] | 0. 25[c] | 0. 53[g] | 1445[h] | 5.76[c] |
| Sesame | 2.03[c] | 0. 41[c] | 55. 13[c] | 3. 9 [b] | 2. 11[c] | 8. 64[c] | 0. 25[c] | 0. 53[g] | 1445[h] | 5.76[c] |
| Sugar beet | 2. 03[c] | 0. 41[c] | 55. 13[c] | 3. 9[b] | 2. 11[c] | 8. 64[c] | 0. 25[c] | 0. 53[g] | 1445[h] | 5.76[c] |
| Coniferous forest | 2.65[j] | 0.11[j] | 118[e] | 6[e] | 2.4[e] | 28[e] | 1[i] | 3.5[e] | 1514[e] | 9.7[e] |
| Broadleaf forest | 1.181[*] | 0.31[e] | 102[e] | 5[e] | 1.3[e] | 14[e] | 1[i] | 1.5[e] | 1630[e] | 13[e] |
| Mixed forest | 9.2[e] | 0.6[e] | 102[e] | 5[e] | 1.3[e] | 14[e] | 1[i] | 1.5[e] | 1630[e] | 9.7[e] |
| Grassland | 2.6[e] | 0.4[e] | 59[e] | 1.5[e] | 2.8[e] | 9.3[e] | 0.5[e] | 0.5[e] | 1692[e] | 5.4[e] |
| Shrubland | 6.6[e] | 0.5[e] | 68[e] | 2.6[e] | 3.9[e] | 4.8[e] | 0.7[e] | 1.2[e] | 1716[e] | 9.3[e] |

References: [a] Cao et al. (2008); [b] Li et al. (2007); [c] He et al. (2015); [d] Tang et al. (2014); [e] Akagi et al. (2011); [f] Zhang et al. (2009); [g] EPD (2014); [h] Wang et al. (2008); [i] Andreae and Rosenfeld (2008); [*] This study.



**Table 6.** Cumulative emissions of major pollutants from open biomass burning in Central and Eastern China during 2003-2015 (Gg yr$^{-1}$)

| Province | OC | EC | CH$_4$ | NOx | NMVOCs | SO$_2$ | NH$_3$ | CO | CO$_2$ | PM$_{2.5}$ |
|---|---|---|---|---|---|---|---|---|---|---|
| Shandong | 783.9 | 48.56 | 669.4 | 479.3 | 1505 | 54.55 | 95.56 | 10880 | 226705 | 1007 |
| Henan | 1068 | 63.19 | 738.3 | 512.1 | 1629 | 54.23 | 101.3 | 11869 | 260239 | 1155 |
| Anhui | 238.2 | 20.24 | 197.7 | 115 | 410 | 12.94 | 29.75 | 2939 | 63623 | 283.1 |
| Jiangsu | 201.6 | 19.88 | 178 | 98.48 | 341 | 9.29 | 23.89 | 2543 | 53106 | 228.5 |
| Hubei | 234.2 | 33.92 | 337.7 | 173.1 | 660.7 | 19.86 | 48.5 | 4555 | 97788 | 415.8 |
| Hunan | 202 | 40.34 | 376.8 | 179.1 | 738.4 | 24.33 | 64.3 | 5239 | 96338 | 418.8 |
| Jiangxi | 132.8 | 27.88 | 236.1 | 109 | 447.6 | 14.2 | 40.55 | 3305 | 57692 | 252.3 |
| Fujian | 97.15 | 15.15 | 148.1 | 71.14 | 347.4 | 12.81 | 34.45 | 2285 | 40095 | 190.2 |
| Zhejiang | 91.41 | 16.22 | 147.9 | 70.53 | 290.9 | 9.62 | 25.83 | 2055 | 39142 | 167.8 |
| Shanghai | 14.34 | 2.09 | 17.14 | 8.56 | 29.89 | 0.76 | 2.29 | 233.8 | 4392 | 17.88 |
| Total | 3064 | 287.5 | 3047 | 1816 | 6399 | 212.6 | 466.5 | 45904 | 939120 | 4136 |

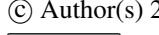


**Table 7.** Correction analysis of the variation tendency from 2003 to 2015 between PM$_{2.5}$ emission from crops straw burning in each province and the rural population, agricultural output and per capita incomes of rural residents.

| PM$_{2.5}$ emission (Gg) | Rural population (10 thousand) | Per capita income of rural residents (RMB) | Agricultural output (0.1 billion RMB) |
|---|---|---|---|
| Shanghai | y = -0.001x + 1.64<br>R$^2$ = 0.17 P > 0.05 | y = -5E-06x + 1.4<br>R$^2$ = 0.09 P > 0.05 | y = 7E-05x + 1.36<br>R$^2$ = 0.0005 P > 0.05 |
| Zhejiang | y = 0.002x + 6.19<br>R$^2$ = 0.06 P > 0.05 | y = -6E-05x + 10.47<br>R$^2$ = 0.19 P > 0.05 | y = -0.001x + 10.72<br>R$^2$ = 0.19 P > 0.05 |
| Fujian | y = -0.0002x + 8.219<br>R$^2$ = 0.01 P > 0.05 | y = -3E-05x + 8.1884<br>R$^2$ = 0.06 P > 0.05 | y = -0.0002x + 8.2144<br>R$^2$ = 0.06 P > 0.05 |
| **Jiangsu** | **y = -0.002x + 23.41**<br>**R$^2$ = 0.8 P < 0.01** | **y = 0.0002x + 15.33**<br>**R$^2$ = 0.66 P < 0.01** | **y = 0.001x + 15.18**<br>**R$^2$ = 0.69 P < 0.01** |
| **Hubei** | **y = -0.008x + 56.19**<br>**R$^2$ = 0.94 P < 0.01** | **y = 0.0009x + 25.39**<br>**R$^2$ = 0.86 P < 0.01** | **y = 0.004x + 24.31**<br>**R$^2$ = 0.92 P < 0.01** |
| **Anhui** | **y = -0.005x + 37.11**<br>**R$^2$ = 0.91 P < 0.01** | **y = 0.0007x + 16.12**<br>**R$^2$ = 0.79 P < 0.01** | **y = 0.004x + 14.5**<br>**R$^2$ = 0.85 P < 0.01** |
| **Hunan** | **y = -0.01x + 62.66**<br>**R$^2$ = 0.78 P < 0.01** | **y = 0.0008x + 20.66**<br>**R$^2$ = 0.8 P < 0.01** | **y = 0.003x + 20.1**<br>**R$^2$ = 0.91 P < 0.01** |
| **Jiangxi** | **y = -0.008x + 33.73**<br>**R$^2$ = 0.92 P < 0.01** | **y = 0.0006x + 11.19**<br>**R$^2$ = 0.82 P < 0.01** | **y = 0.006x + 9.84**<br>**R$^2$ = 0.87 P < 0.01** |
| **Henan** | **y = -0.01x + 150.14**<br>**R$^2$ = 0.8 P < 0.01** | **y = 0.003x + 70.41**<br>**R$^2$ = 0.59 P < 0.01** | **y = 0.008x + 62.79**<br>**R$^2$ = 0.72 P < 0.01** |
| **Shandong** | **y = -0.009x + 122.46**<br>**R$^2$ = 0.73   P < 0.01** | **y = 0.0014x + 66.48**<br>**R$^2$ = 0.66 P < 0.01** | **y = 0.004x + 62.11**<br>**R$^2$ = 0.77 P < 0.01** |





**Table 8.** Comparison of the emissions with previous studies in different years (Gg yr$^{-1}$)

| Reference | Year | OC | EC | CH$_4$ | NO$_x$ | NMVOCs | SO$_2$ | NH$_3$ | CO | CO$_2$ | PM$_{2.5}$ |
|---|---|---|---|---|---|---|---|---|---|---|---|
| Wang et al., 2008 | 2006 | 252 | 25.8 | 197 | 189 | 459 | 31.8 | 44.1 | 3841 | 81225 | 1138 |
| This study | | 215.3 | 21.13 | 3267 | 220.7 | 131.9 | 451.1 | 14.33 | 31.46 | 67753 | 293.09 |
| Huang et al., 2012 | 2006 | 54 | 17.4 | 136 | 123 | 1196 | 8.1 | 50.6 | 2379 | 36886 | 146 |
| This study | | 209.8 | 20.67 | 215.8 | 129.1 | 436.4 | 13.56 | 29.64 | 3172 | 66088 | 283.3 |
| Qiu et al., 2016 | 2013 | 222 | 41.5 | 243 | 168 | 591 | 30.2 | 46.9 | 3273 | 78633 | 475 |
| This study | | 258.2 | 23.53 | 3817 | 252.1 | 151.2 | 531.5 | 17.86 | 38.67 | 78050 | 343.44 |
| Zhou et al., 2017 | 2012 | 185 | 16.9 | 254 | 160 | 543 | 40.4 | 34.5 | 3330 | 92797 | 484 |
| This study | | 248.6 | 23.11 | 3688 | 245.7 | 148.5 | 507.8 | 16.71 | 35.92 | 75785 | 329.46 |



**Table 9.** The uncertainty estimation of open biomass burning emissions for various pollutants from 2003 to 2015.

| Year | OC | EC | CO | CH$_4$ | NO$_X$ | NMVOC | SO$_2$ | NH$_3$ | CO$_2$ | PM$_{2.5}$ |
|---|---|---|---|---|---|---|---|---|---|---|
| 2003 | (-31%, 31%) | (-46%, 46%) | (-20%, 20%) | (-20%, 20%) | (-23%, 23%) | (-52%, 53% | (-52%, 51%) | (-33%, 33%) | (-3%, 3%) | (-44%, 44%) |
| 2004 | (-29%, 29%) | (-47%, 48%) | (-21%, 21%) | (-22%, 22%) | (-24%, 24%) | (-45%, 45%) | (-56%, 58%) | (-34%, 34%) | (-3%, 3%) | (-47%, 47%) |
| 2005 | (-31%, 31%) | (-42%, 44%) | (-16%, 16%) | (-16%, 17%) | (-19%, 19%) | (-41%, 40%) | (-44%, 44%) | (-32%, 33%) | (2%, 3%) | (-35%, 34%) |
| 2006 | (-32%, 33%) | (-44%, 44%) | (-13%, 13%) | (-14%, 14%) | (-16%, 17%) | (-43%, 43%) | (-34%, 35%) | (-34%, 34%) | (-3%, 3%) | (-25%, 25%) |
| 2007 | (-30%, 30%) | (-46%, 46%) | (-18%, 19%) | (-19%, 19%) | (-22%, 22%) | (-50%, 51%) | (-50%, 50%) | (-33%, 34%) | (-3%, 3%) | (-42%, 42%) |
| 2008 | (-26%, 26%) | (-52%, 53%) | (-25%, 25%) | (-28%, 28%) | (-29%, 29%) | (-69%, 69%) | (-62%, 61%) | (-38%, 39%) | (-3%, 3%) | (-55 %, 56%) |
| 2009 | (-28%, 28%) | (-48%, 48%) | (-21%, 21%) | (-21%, 22%) | (-24%, 24%) | (-59%, 59%) | (-54%, 54%) | (-34%, 35%) | (-3%, 3%) | (-47%, 47%) |
| 2010 | (-31%, 31%) | (-44%, 44%) | (-16%, 17%) | (-17%, 17%) | (-19%, 19%) | (-45%, 46%) | (-42%, 42%) | (-33%, 34%) | (-3%, 3%) | (-34%, 34%) |
| 2011 | (-29%, 29%) | (-46%, 46%) | (-18%, 18%) | (-19%, 19%) | (-21%, 21%) | (-52%, 53%) | (-47%, 47%) | (-34%, 35%) | (-3%, 3%) | (-40%, 40%) |
| 2012 | (-32%, 33%) | (-44%, 44%) | (-14%, 14%) | (-14%, 14%) | (-17%, 17%) | (-35%, 35%) | (-35%, 35%) | (-34%, 35%) | (-3%, 3%) | (-27%, 26%) |
| 2013 | (-30%, 30%) | (-44%, 44%) | (-16%, 16%) | (-17%, 17%) | (-20%, 20%) | (-51%, 51%) | (-42%, 43%) | (-33%, 34%) | (-3%, 3%) | (-36%, 36%) |
| 2014 | (-32%, 32%) | (-45%, 46%) | (-15%, 15%) | (-16%, 16%) | (-19%, 18%) | (-43%, 43%) | (-42%, 42%) | (-35%, 35%) | (-3%, 3%) | (-33%, 33%) |
| 2015 | (-31%, 31%) | (-44%, 44%) | (-14%, 146%) | (-14%, 13%) | (-17%, 17%) | (-41%, 41%) | (-34%, 34%) | (-34%, 35%) | (-3%, 3%) | (-26%, 26%) |



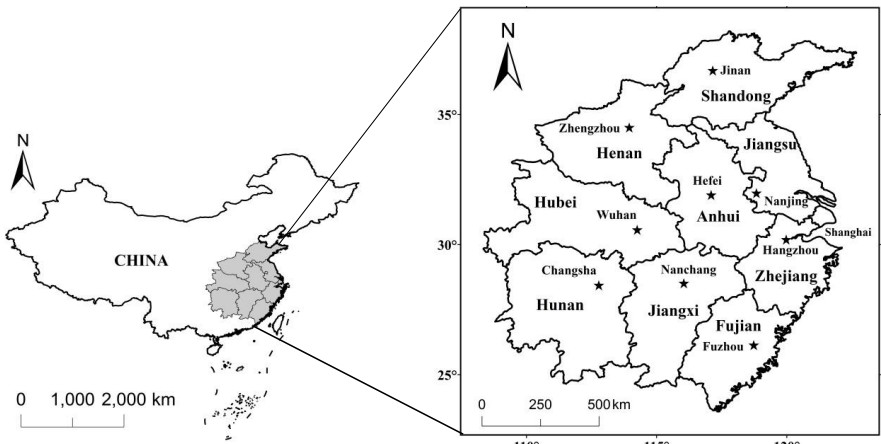

**Figure 1.** Location of Central and Eastern China and the key megacities.

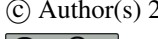



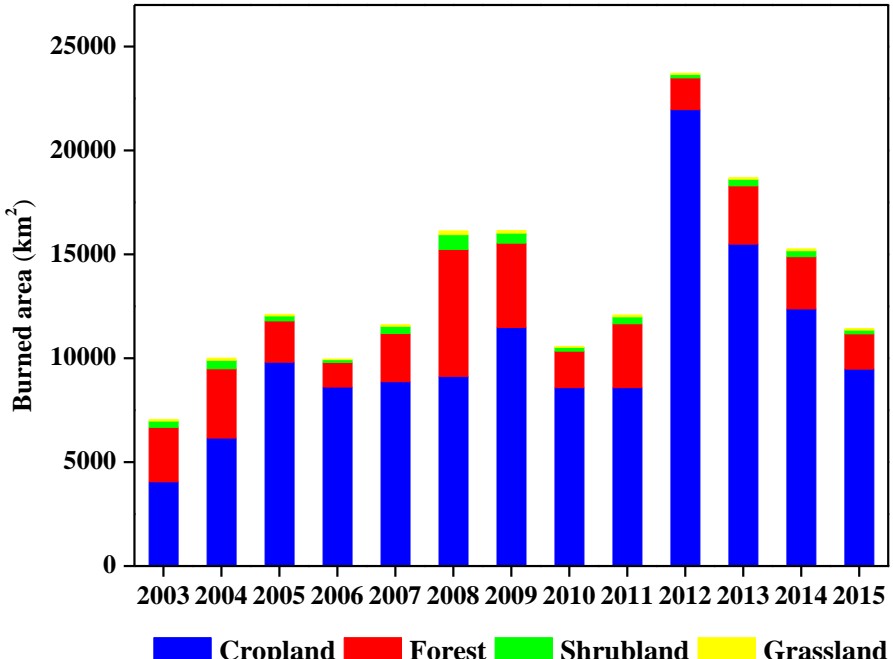

**Figure 2.** The integrated open biomass burned area in Central and Eastern China from 2003 to 2015.



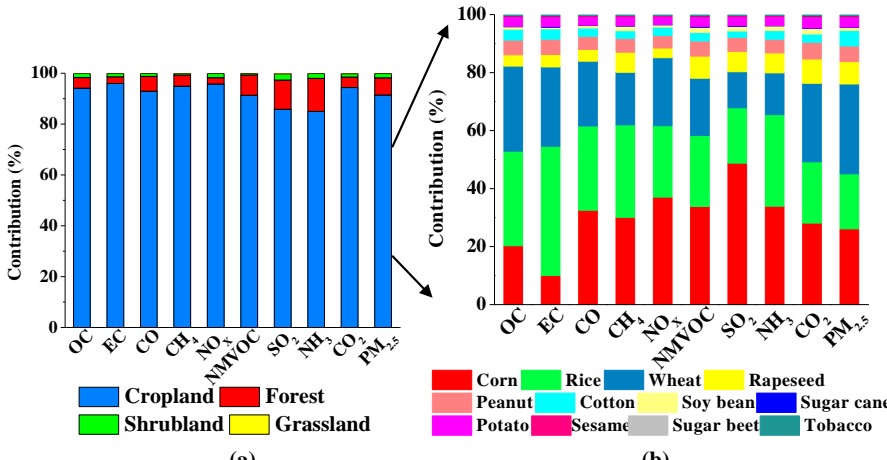

**Figure 3.** The mean contributions of different types of biomass (a) and the contribution of different types of crops to the whole cropland accumulative emissions of pollutants in Central and Eastern China from 2003-2015.





**Figure 4.** The averaged contributions of different biomass burning types to PM$_{2.5}$ emission in each province.





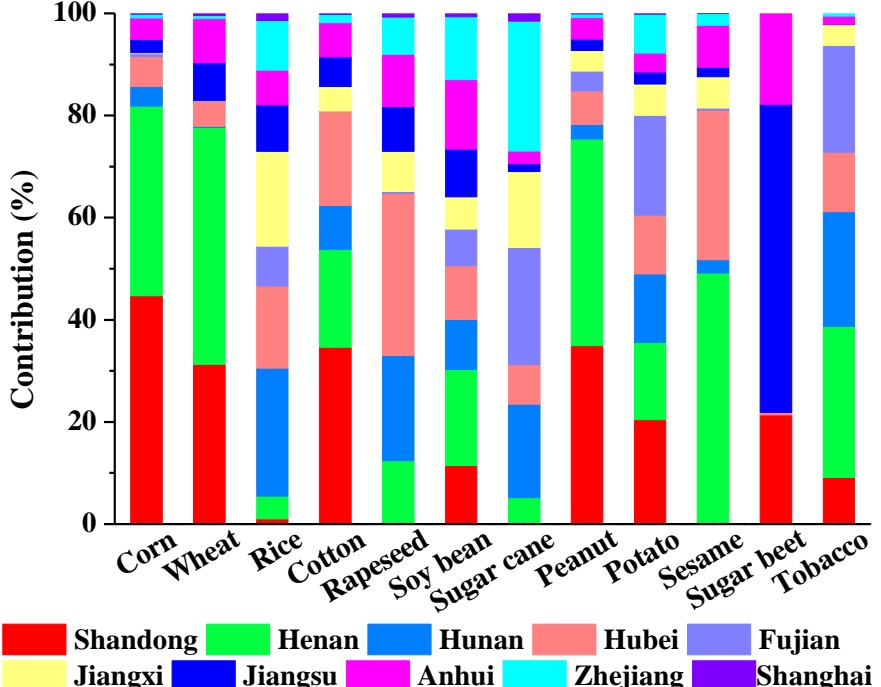

**Figure 5.** The averaged contributions of various crops straw burning to PM$_{2.5}$ emission in different provinces.




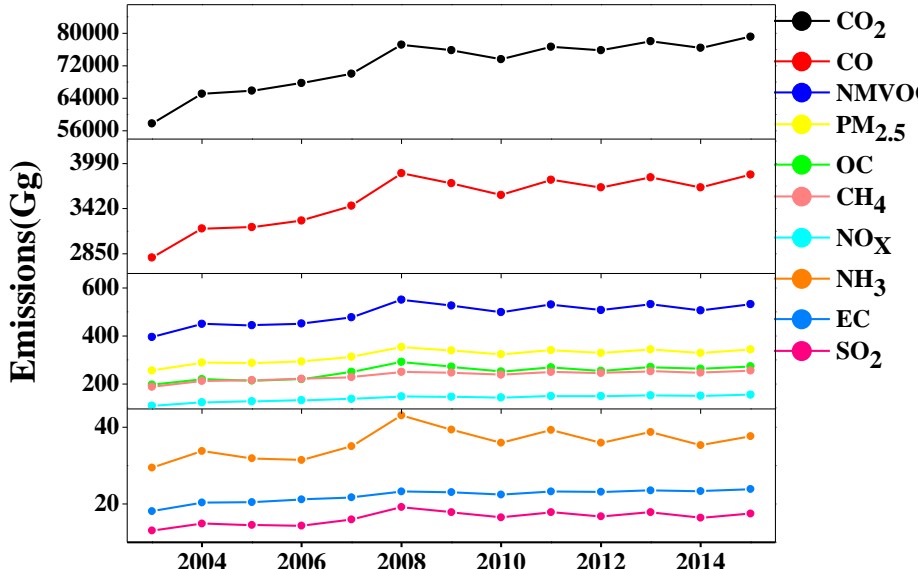

**Figure 6.** Historical emissions of opening biomass burning from 2003 to 2015.

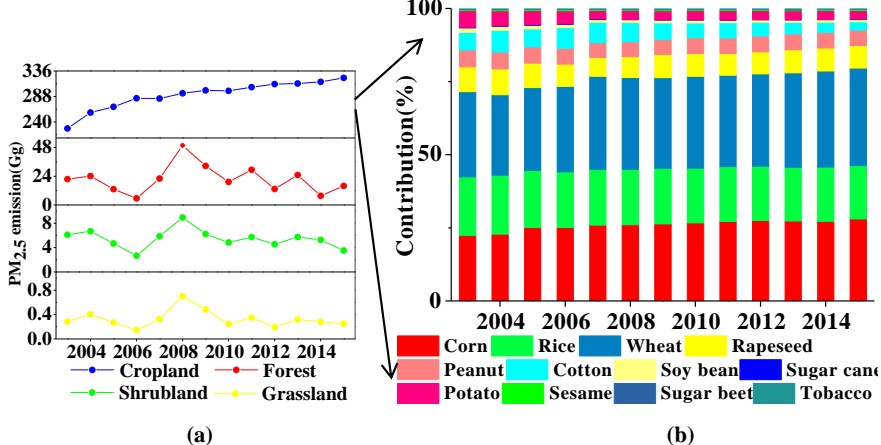

**Figure 7.** The multi-year PM$_{2.5}$ emissions from (a): different opening biomass burning sources; (b): various crop types from 2003 to 2015.



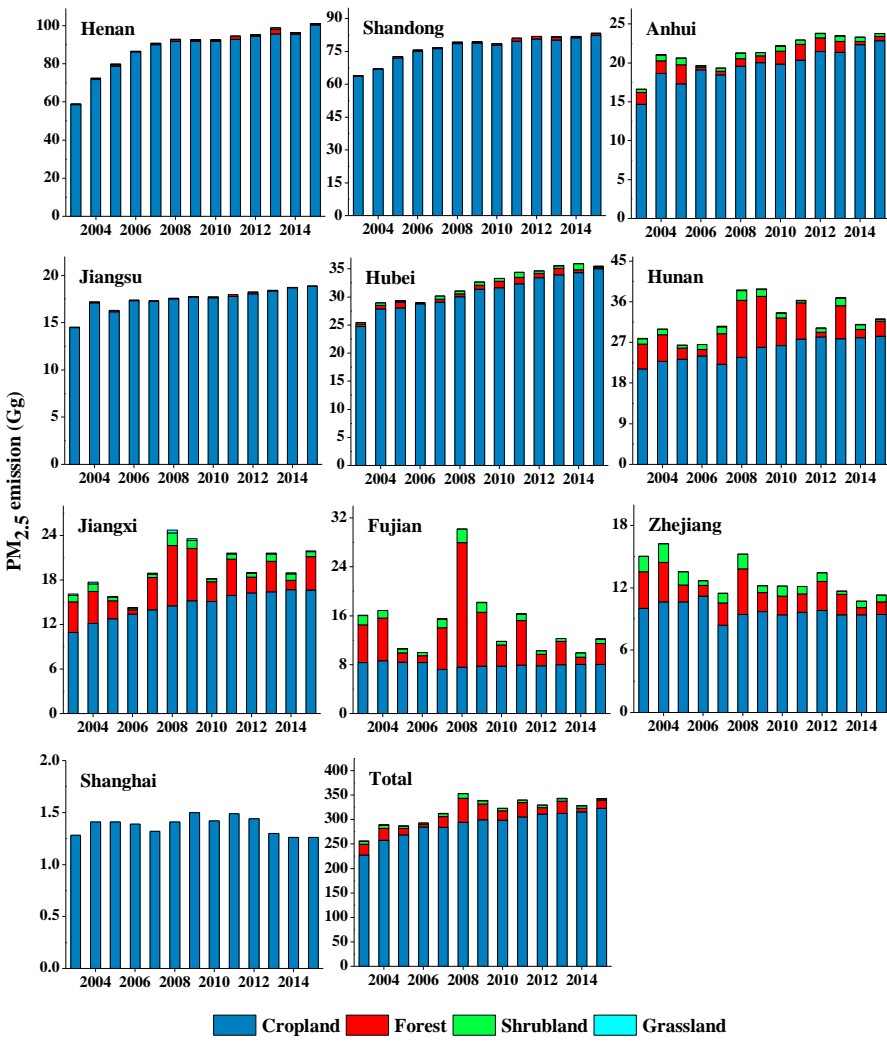

**Figure 8.** The multi-year PM$_{2.5}$ emission for the four types of biomass burning in different provinces from 2003 to 2015.





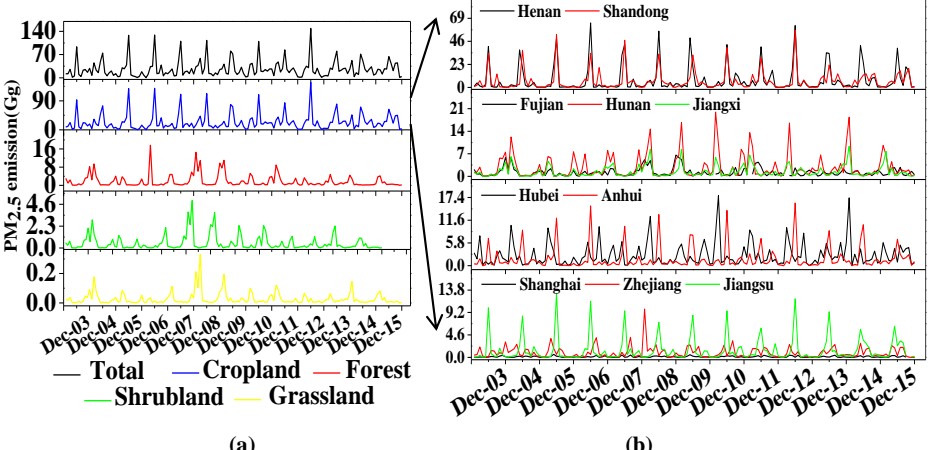

**Figure 9.** The monthly PM$_{2.5}$ emission of different open biomass burning from 2003 to 2015 for
the whole Central and Eastern China (a) and for each province (b).





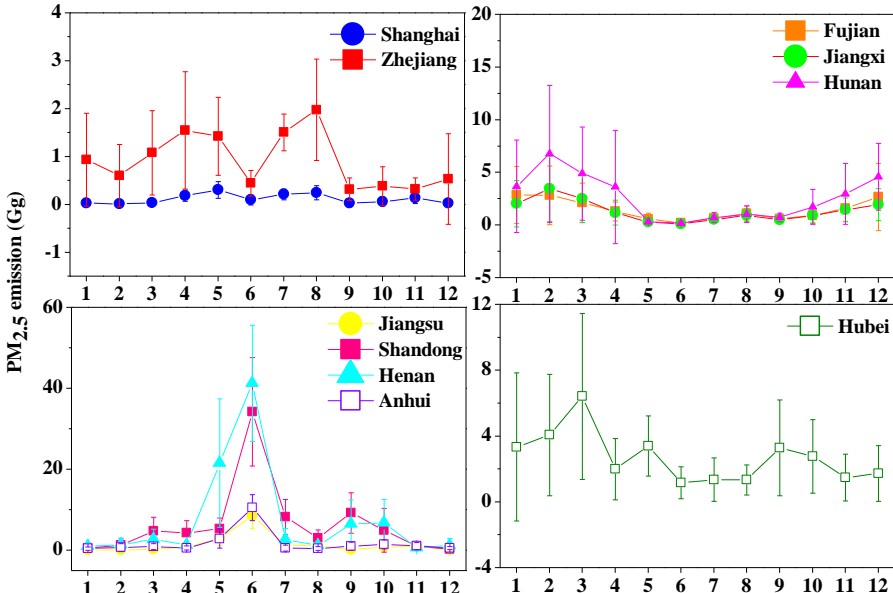

**Figure 10.** The Monthly PM$_{2.5}$ emission from open biomass burning in each province.



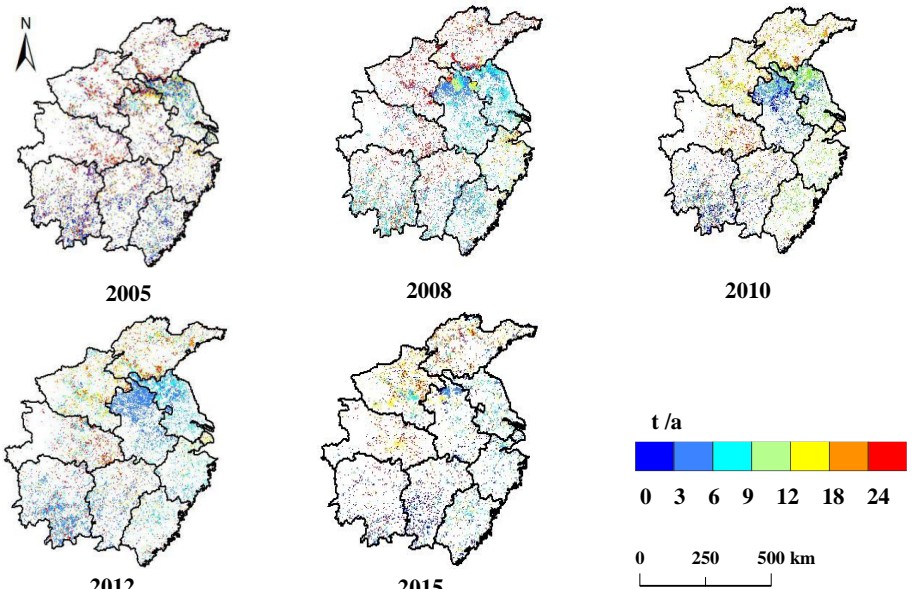

**Figure 11.** Annual spatial distribution (1 km×1 km) of PM$_{2.5}$ emissions from opening biomass burning in Central and Eastern China.





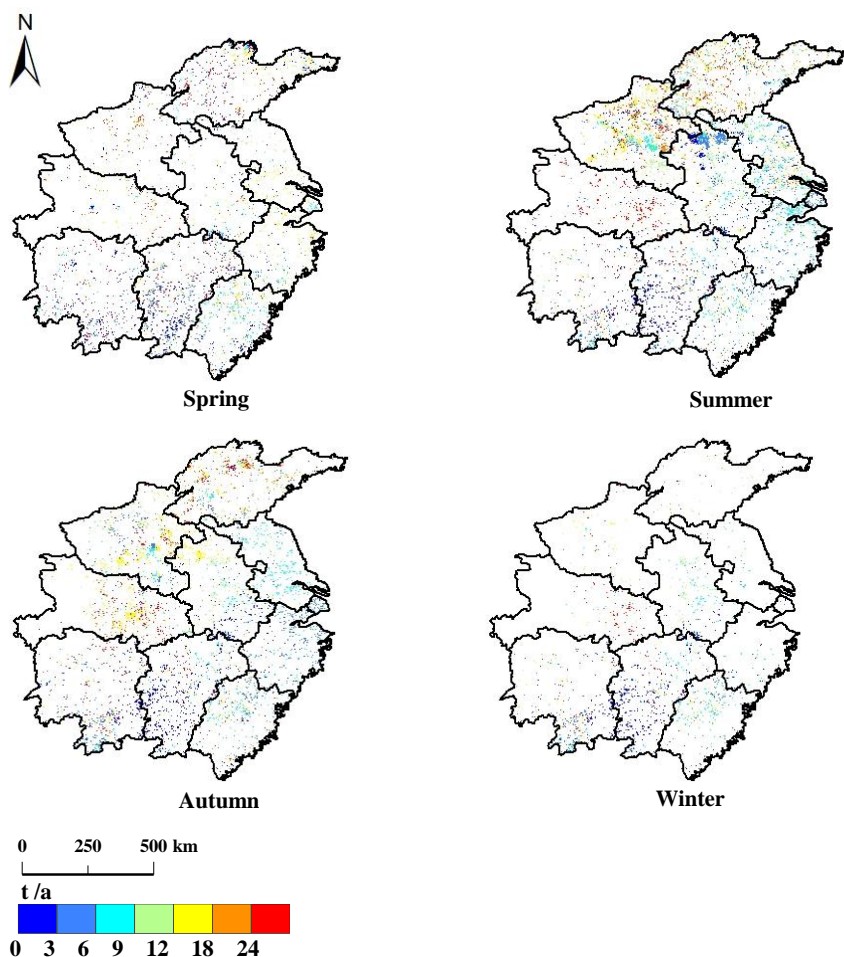

**Figure 12.** Seasonal emission distribution (1 km×1 km) of PM$_{2.5}$ in 2015 from opening biomass burning in Central and Eastern China.



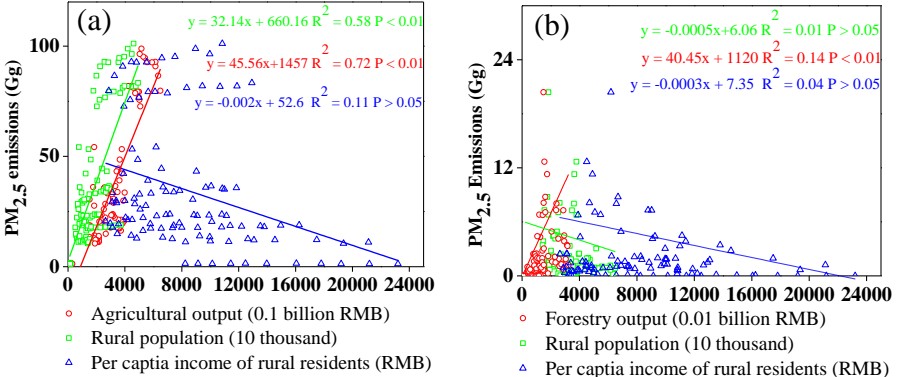

**Figure 13.** Correction analysis between PM$_{2.5}$ emission from (a) crop residue burning and (b) forestry fire burning in different provinces and agricultural output and forestry output, rural population and per captia incomes of rural residents from 2003-2015.