# Peer review of "Estimating the open biomass burning emissions in Central and Eastern China from 2003 to 2015 based on satellite observation"

_Atmospheric Chemistry and Physics, 2018_

## Referee Comment (RC1) · Anonymous Referee #1 · 2 May 2018

This manuscript discussed the open biomass burning emissions in Central and Eastern China including several provinces with different vegetation and also the emissions in different years and seasons. research. The authors also estimated the pollutions from open biomass burning in this area from 2003 to 2015. The spatial and temporal distribution of open biomass burning provides a high resolution result to relevant researchers and could be meaningful in policy making. And there are some technical questions that need the authors to clarify (see additional comments).

Additional comments: 1. Line 27: The initialism needs to be explained for the first use, i.e. OC, EC and NMVOC.

[Figure]

2. Line 100-101: "". Please review the structure of this sentence.

3. Section 2.3: Although the author set different CE values for different vegetation, the CE was set as a constant during each open burning process. Please discuss about it and provide a reasonable explanation, because the CE should not be a constant during burning and the pollution emissions were not uniform in different phase.

4. Line 219-221: The contribution presented in this part were with different significant figures or decimal digits. Please explain why you use different significant figures or decimal digits for different contributors?

5. Line 257: Please explain the increasing trend of OBB emission from 2003 to 2008. It seems that the explanation in Line 269-280 was not convincing enough.

6. Line 452-460: Please specify how much do your research improve the uncertainties in OBB emission and contribution estimation.

---

## Referee Comment (RC2) · Anonymous Referee #2 · 21 May 2018

The manuscript by Wu et al. presents new estimates of emissions from open biomass burning over Central and Eastern China. Its novelty lies in the fact that it blends various sources of information on burnt area (from satellites), biomass availability, emission factors, and socioeconomic aspects to produce emissions estimates that are of high resolution and consistent with recent knowledge. The authors also go some way towards comparing their estimates with previous studies and quantifying the uncertainty of their estimates, though the latter analysis could have been somewhat more thorough and informative (see comment below). The study is within the remit of Atmospheric Chemistry and Physics and I find the results worthy of publication, after some improvements that I describe below.

[Figure]

GENERAL COMMENTS:

- The uncertainty discussion in Sect. 3.5 can be improved. Presumably for the Monte Carlo simulations, some uncertainty range had to be selected for each individual variable involved in the emissions calculation, but it is not clear what this range is for some variables, such as for example burned area. Also, the authors do not explain what is it that makes specific pollutants (e.g. EC) have larger uncertainty and others smaller? Which of the variables needed for calculating emissions have the biggest role in driving uncertainty? This will help focus future efforts in order to reduce uncertainty in emissions in this area. Finally, when the authors say that "Compared with previous studies, for the emission estimation of forest burning, the uncertainty was improved" or "For cropland, the uncertainty was improved. . .", do they mean that the other studies estimated uncertainty ranges which were wider, or that the datasets used here are better/more suitable?

- The use of English can be polished throughout the manuscript. Not that the language has major problems – more or less it is fine. But almost in every sentence (or every other sentence) I found myself thinking that the choice of words, grammar or syntax could be improved. I have made some suggestions, but there are more places where things can be improved.

SPECIFIC COMMENTS:

Line 19: Please change "few focus" to "little focus".

Line 33: Is "per pixel" informative for the average reader?

Line 35: Please change "for" to "from".

Lines 41-43: I think that these sentences would be better suited for the beginning of the abstract, rather than the end.

Line 84: "accurate" -> "accurately"

Line 114: "the same emissions factors": The same to what?

Line 119: "historical" might be too strong a word here. Suggest using "of recent years" or something similar.

Line 131: Shouldn't "E" also be indexed with x and t?

Line 134: Mention explicitly that "CE" stands for combustion efficiency. Also, CE does not have a subscript x in the equation, whereas it does in this line.

Line 147: Is v the same as j in the earlier equation? Then why change the symbol? Same lower down for r and s.

Line 148: Not clear (to me at least) what "outside of the burned area" means here. I think generally Equation 2 would benefit from a somewhat clearer explanation for the readers not familiar with the technical details of the products.

Lines 154-158: So, is the biomass basically a step function? Is this justifiable? Is this expected to generate any artifacts in the results?

Lines 188-189: What does "similar" mean here?

Line 190: Is there a reference for this previous research?

Line 193: "opening" -> "open"

Line 202: Maybe the title should be "Other factors influencing OBB emission"?

Line 209: "20, 000" -> "20,000"

Lines 214-215: It should be made clear somewhere that PM2.5 and OC/EC are not totally different things, i.e. PM will include some OC and EC.

Line 217: Suggest removing "section".

Line 240: "This is mostly": Which? For example, what are the "suitable weather conditions" in this case?

Figure 6: "opening" -> "open"

Lines 255-259: There is a confusion between trends and short-term (interannual) variations here. Can the authors perhaps provide the overall trend for each species for the whole period, and whether it is statistically significant?

Line 265: Presumably it is just coincidental that they are odd and even years. Therefore maybe not worth the emphasis?

Line 280: Increased or decreased?

Line 288: "scattering" -> "scattered"

Line 315: "totally occupied. . .": Please improve phrasing.

Line 318: "based on the correlation of emissions in each month" -> "based on the correlation between their monthly emissions"

Line 328: "occurred" -> "occurring"

Line 345: "were found" -> "featured"

Lines 348-349: "are uniformed" -> "remain similar"

Line 378: May need to improve terminology a bit. For example, why is "local burning habits" not part of "anthropogenic activities"? What is the distinction?

Line 380: "People sweep their graves": The authors may want to change the wording as "their" might be a bit misleading here...

Line 382: "file" -> "fire"

Line 406: "where" -> "which"

Line 416: "have" -> "having"

Line 420: Maybe "close" is a bit of an overstatement? They should not be expected to be very close anyway as there seem to have been several improvements in the current

study.

Line 421-422: "The differences were mainly caused...": How do the authors know that these are the causes of differences? And which of the factors may be more responsible?

Line 431: Are lower EFs used here more justifiable?

Lines 436-439: This can be combined into one sentence to avoid repetition.

Line 438: "localized in CEC" -> "specific to CEC"

Line 439: "can improve" -> "are likely to have improved"

Line 446: Do 0.03 and 0.85 refer to relative uncertainties?

Line 446: "At last" -> "Finally"

Line 474: "opening" -> "open"

Line 487: "of the next year" is not needed.

Line 488: "impacted on the emission": This implies causality but the analysis has been based on correlations, which can only be suggestive of possible associations but not conclusive for cause-effect relationships.
* * *

---

## Author Comment (AC1) · 19 Jul 2018

**Authors' Responses to Reviewer Comments**

Manuscript: Estimating the open biomass burning emissions in Central and Eastern China from 2003 to 2015 based on satellite observation (Ref. No.: acp-2018-282)

We are very grateful for your careful and insightful comments, which contributed greatly to improve this manuscript. We carefully answered them point-by-point as below and corrected the corresponding parts in the manuscript (in red color).

This manuscript discussed the open biomass burning emissions in Central and Eastern China including several provinces with different vegetation and also the emissions in different years and seasons. The authors also estimated the pollutions from open biomass burning in this area from 2003 to 2015. The spatial and temporal distribution of open biomass burning provides a high resolution result to relevant researchers and could be meaningful in policy making. And there are some technical questions that need the authors to clarify (see additional comments).

Response:

Thanks a lot for your positive comments on this manuscript. We have improved this manuscript by answering the reviewers' comments and advice.

Additional comments:

1. Line 27: The initialism needs to be explained for the first use, i.e. OC, EC and NMVOC.

Response:

Thanks for this suggestion. We have corrected it in the manuscript (in Line 30-32): "...organic carbon (OC), elemental carbon (EC), methane ($CH_4$), nitric oxide ($NO_X$), non-methane volatile organic compounds (NMVOCs), sulfur dioxide ($SO_2$), ammonia ($NH_3$), carbon monoxide (CO), carbon dioxide ($CO_2$) and fine particles ($PM_{2.5}$)..."

2. Line 100-101: "". Please review the structure of this sentence.

Response:

We have polished it as shown in Line 106-107.

*"One is the burned area product, which provides fire burned areas of the whole month. It is limited by the lower pixel resolution."*

3. Section 2.3: Although the author set different CE values for different vegetation, the CE was set as a constant during each open burning process. Please discuss about it and provide a reasonable explanation, because the CE should not be a constant during burning and the pollution emissions were not uniform in different phase.

Response:

We really appreciate for this comment. We highly agree with you that the CE should not be a constant during burning and the pollution emissions were not uniform in different burning phase. However, the emission inventories in this research and currently published papers were estimated for a long time period or a whole year with the time scale as month, instead of hour. Therefore, the "CE" values used here reflected the average biomass burning condition for the whole open burning processes, which can not reflect the different combustion phase (like smoldering and flaming, etc.).

It is the research hotspot in developing emission inventory with high time resolution, which needs both the high time-resolution activity data and emission factors for different burning stages. It is also now being considered in our research group.

We accepted this suggestion and added corresponding discussion in the manuscript (Line 212-219).

Line 219-221: The contributions presented in this part were with different significant figures or decimal digits. Please explain why you use different significant figures or decimal digits for different contributors?

Response:

Thank you very much for this comment. We are truly sorry for the confusion in the use of different significant figures or decimal digits. We have made the revision in the manuscript, by using the same decimal digits for different contributors (Line 254-256).

4. Line 257: Please explain the increasing trend of OBB emission from 2003 to 2008. It seems that the explanation in Line 269-280 was not convincing enough.

Response:

Thanks for this suggestion. In Figure 6, we found that the emissions of $PM_{2.5}$ from crop burning significantly increased from 2003 (228 Gg) to 2008 (294 Gg), due to the increase of crops production and deficiency of strict control policies in this period (Table S1). Emissions from forest, shrubland and grassland burning exhibited an obvious declining trend from 2003 to 2006 and then increased from 2006 to 2008, which maybe related with the different weather conditions and human forestry activities. Although

emissions from forest, shrubland and grassland burning fluctuated markedly during this period, the obvious increase of crops residue burning dominated the total growth of OBB emission from 2003 to 2008.

Corresponding revision and discussion has been added in the manuscript (Line 323-327).

5. Line 452-460: Please specify how much do your research improve the uncertainties in OBB emission and contribution estimation.

Response:

Thanks for this query. In this study, we improve the uncertainty estimation method based on reliable multiple satellites data. In addition, the uncertainties of active data sets (biomass loading data and local EFs) were improved. Multiple satellites can better obtain burned area data; different active data were more suitable because they could better reflect the actual situation in Central and Eastern China. Meanwhile, the field survey also helped to improve the uncertainty and reliability of our emission results. Through these improvements, the uncertainty of pollutant emission was improved.

As shown in Table 1, the uncertainty ranges of different pollutant emissions were narrowed.

**Table 1.** Uncertainty ranges of different pollutants in emission estimation.

| Pollutant | Uncertainty ranges | | |
|---|---|---|---|
| | Our study | Wang et al., 2008 | Qiu et al., 2016 |
| OC | ±30% | ±148% | -72% - 213% |
| EC | ±48% | ±132% | -67% - 204% |
| $CH_4$ | ±20% | ±77% | -32% - 81% |
| $NO_X$ | ±20% | ±80% | -36% - 92% |
| NMVOCs | ±45% | ±71% | -24% - 78% |
| $SO_2$ | ±45% | ±108% | -47% - 121% |
| $NH_3$ | ±35% | ±137% | -61% - 152% |
| CO | ±18% | ±86% | -52% - 105% |
| $CO_2$ | ±3% | ±60% | |
| $PM_{2.5}$ | ±36% | ±142% | -77% - 213% |

**Response**

Wang, S. X. and Zhang, C. Y.: Spatial and Temporal Distribution of Air Pollutant Emissions from Open

Burning of Crop Residues in China, Science paper Online, 3, 329–333, 2008 (in Chinese).

Qiu, X., Duan, L., Chai, F., Wang, S., Yu, Q. and Wang, S.: Deriving High-Resolution Emission Inventory of OBB in China based on Satellite Observations, Environ. Sci. Technol., 50(21), 11779-11786, doi:10.1021/acs.est.6b02705, 2016.

---

## Author Comment (AC2) · 19 Jul 2018

**Authors' Responses to Reviewer Comments**

Manuscript: Estimating the open biomass burning emissions in Central and Eastern China from 2003 to 2015 based on satellite observation (Ref. No.: acp-2018-282).

Thanks a lot for your positive and constructive comments and suggestions. We carefully checked the issues and answered them point-by-point as below. Corresponding corrections were done in the manuscript (in red color).

General comment:

The uncertainty discussion in Sect. 3.5 can be improved. Presumably for the Monte Carlo simulations, some uncertainty range had to be selected for each individual variable involved in the emissions calculation, but it is not clear what this range is for some variables, such as for example burned area. Also, the authors do not explain what is it that makes specific pollutants (e.g. EC) have larger uncertainty and others smaller? Which of the variables needed for calculating emissions have the biggest role in driving uncertainty? This will help focus future efforts in order to reduce uncertainty in emissions in this area. Finally, when the authors say that "Compared with previous studies, for the emission estimation of forest burning, the uncertainty was improved" or "For cropland, the uncertainty was improved. . .", do they mean that the other studies estimated uncertainty ranges which were wider, or that the datasets used here are better/more suitable?

The use of English can be polished throughout the manuscript. Not that the language has major problems - more or less it is fine. But almost in every sentence (or every other sentence) I found myself thinking that the choice of words, grammar or syntax could be improved. I have made some suggestions, but there are more places where things can be improved.

**General response:**

We really appreciate for your constructive suggestions and comments, which will be helpful for improving this manuscript. We are regretful for the unclear description. The following is an explanation for the parameters used in Monte Carlo simulation.

We assess the emission uncertainty of specific pollutants by the Monte Carlo simulation, which was widely used in uncertainty estimation in previous studies (e.g., Streets et al., 2003; Li et al., 2016). The variables involved in emission calculation included burned area, biomass loading data,

combustion efficiency and emission factors. According to previous researches, the estimation of fires burned area was proved to be reliable by the adoption of burned area product MCD64AL (Giglio et al., 2013) and active fire product MCD14ML (Randerson et al., 2012). It is difficult to assess the uncertainty of the satellite-derived data for the burned land area (Hoelzemann et al., 2004, Chang et al., 2010). Although some active fires which burned out at 10:30 am-1:30 pm each day could not be captured by MCD14ML, the burned area used in this study were more reliable due to the combination of multiple satellite datasets (MCD64AL and MCD14ML). The uncertainties in this study were mainly caused by biomass loading data, combustion efficiency and emission factors. These data were assumed to be normal distribution (Zhao et al., 2011). The uncertainty of biomass loading data and combustion efficiency was estimated with a standard deviation of approximately 50% of the mean value (Shi et al., 2015). The uncertainty for emission factors was cited from references as listed in Table S5, with the uncertainty for each pollutant mainly ranged from 0.03 to 0.85. The reliable of emission factors play the most important role in driving uncertainty. Considering all these parameters, 20,000 Monte Carlo simulations were performed to evaluate the estimation uncertainty quantitatively for pollutant emissions with 95% coincidence level. We have made corresponding revision in the manuscript (Line 512-526). In addition, Table S5 has also been added in the supplementary file.

Compared with previous studies, the uncertainties were improved in our study due to the datasets used here were better and more suitable: multiple satellites can obtain more reliable and detailed burned area data. Different active data (biomass loading data and local EFs) involved in emission estimation were more suitable because they could better reflect the actual situation in the research region. Overall, the uncertainty ranges of different pollutant emissions were narrowed and more reliable, which could better reflect the real emission.

In addition, we have improved the choice of words, grammar or syntax in the revised manuscript. All the authors have revised the manuscript for another time independently.

We have carefully taken the reviewer's suggestion into consideration during the revision of our paper. Please see the following point-by-point responses.

Specific Comments:
Line 19: Please change "few focus" to "little focus".

**Response:**

It has been corrected (in Line 19).

Line 33: Is "per pixel" informative for the average reader?

**Response:**

Thanks for this query. In this study, all the data were relocated into a 1 km×1 km grid to identify and estimate spatial variation of open biomass burning emission. "per pixel" in line 33 was the same as 1 km×1 km. It has been changed into "per square kilometers" (Line 37-38).

Line 35: Please change "for" to "from".

**Response:**

It has been deleted (Line 40).

Lines 41-43: I think that these sentences would be better suited for the beginning of the abstract, rather than the end.

**Response:**

We accepted this suggestion. The sentences have been moved in Line 23-26.

Line 84:"accurate"-"accurately"

**Response:**

It has been corrected (Line 90).

Line 114: "the same emissions factors": The same to what?

**Response:**

Thank you very much for this comment. We are truly sorry for the unclear description. It means the same emission factors for different pollutants emitted from open biomass burning. We have made the corresponding revision in the manuscript (Line 121).

*"... have used the same emission factors for pollutants emitted from OBB without considering ..."*

Line 119: "historical" might be too strong a word here. Suggest using "of recent years" or something similar.

**Response:**

Thanks for your suggestion. We have corrected the sentence as *"…estimate multi-year OBB emissions from 2003 to 2015…"* (Line 126)

Line 131: Shouldn't "E" also be indexed with x and t?

**Response:**

Sorry for this error. We have changed the "$E_i$" to "$E_{i,x,t}$" in the manuscript. In addition, the description of "$E_{i,x,t}$" have been changed to "emission amount of different pollutants in location x and time t" in the revised manuscript (Line 139-140).

Line 134: Mention explicitly that "CE" stands for combustion efficiency. Also, CE does not have a subscript x in the equation, whereas it does in this line.

**Response:**

We have changed the "CE" into "$CE_x$". The description of "$CE_x$" have been changed to "the combustion efficiency of open biomass burning in location x" in the manuscript (Line 141).

Line 147: Is v the same as j in the earlier equation? Then why change the symbol? Same lower down for r and s.

**Response:**

Sorry for the confusing description. In fact, "v" in the equation is the same as "j" in the earlier equation. We have made the corresponding revision in the manuscript (Line 162).

Considering the difference of definitions, "r" and "s" are not replaced by "i" and "t" in equation 2 in the manuscript.

Line 148: Not clear (to me at least) what "outside of the burned area" means here. I think generally Equation 2 would benefit from a somewhat clearer explanation for the readers not familiar with the technical details of the products.

**Response:**

Thanks for this suggestion.

In this study, we re-sampled the two fire products data into a 1 km×1 km grid. The total burned area in each grid cell was estimated by the following equation (Randerson et al., 2012).

$$BA_{total(i,t,j)} = BA_{MCD64AL(i,t,j)} + BA_{sf(i,t,j)} \tag{1}$$

where $BA_{total(i,t,j)}$ is the total fire burned area in grid cell i, month t and aggregated vegetation class j; $BA_{MCD64AL(i,t,j)}$ is the MCD64AL burned area in grid cell i, month t and aggregated vegetation class j; $BA_{sf(i,t,j)}$ is the small fire burned area in grid cell i, month t and aggregated vegetation class j.

$BA_{MCD64AL(i,t,v)}$ was directly detected from MCD64AL product. MCD14ML active fire points in each grid included two parts: active fires points with or near the MCD64A1 burned area ($FC_{in}$) and active fires outside the MCD64AL burning area ($FC_{out}$). $BA_{sf(i,t,j)}$ was the burned area of $FC_{out}$. The $BA_{sf(i,t,j)}$ was used as supplement.

MCD14ML data could only give the fire points data, but not for the burned area data. Therefore, we used equation 2 to calculate $BA_{sf(i,t,j)}$:

$$BA_{sf(i,t,j)} = FC_{out(i,t,j)} \times \alpha_{(r,s,j)} \times \gamma_{(r,s,j)} \tag{2}$$

Where $BA_{sf(i,t,j)}$ is the small fire burned area of $FC_{out}$ in grid cell i, month t, and aggregated vegetation class j; $FC_{out(i,t,j)}$ is the MCD14 ML active fires outside of the burned area in grid cell i, month t and aggregated vegetation class j; $\alpha_{(r,s,j)}$ and $\gamma_{(r,s,j)}$ are set as coefficients of burned area calculation in burning region r and burning period s, for biomass species j. $\alpha$ is allocated as the ratio of $BA_{MCD64A1}$ to $FC_{in}$ in each grid. It had units of km² per active fire and was used to estimate the burned states of $FC_{in}$ in each grid cell. $\gamma$ is an additional unit less scalar which is used to estimate the difference between $F_{in}$ and $F_{out}$. According to Randerson et al. (2012), $\gamma$ is assumed equal to 1 in China.

We have made the revision correspondingly in the manuscript (Line 150-158).

Lines 154-158: So, is the biomass basically a step function? Is this justifiable? Is this expected to generate any artifacts in the results?

**Response:**

Thanks for this query. It is not just a step function.

The forest biomass density data was defined as the ratio of the total biomass to the total area of different forest species (broadleaf forest, coniferous forest and mixed forest):

$$B_{i,r}=T_{i,r}/A_{i,r} \tag{3}$$

where i stands for different forest species; r means different provinces; $B_{i,r}$ is the biomass density of forest species i in province r; $T_{i,r}$ means the total biomass of forest specie i in province r; $A_{i,r}$ denotes the total area of forest species i in province r.

To obtain $B_{i,r}$, we need to calculated $T_{i,r}$ and $A_{i,r}$, respectively.

$A_{i,r}$ was collected from the 8th Chinese National Forest Resource Inventory (Xu, 2014).

Since different forest species are consist of various tree types, we obtain $T_{i,r}$ by calculating the biomass of each tree types (Fang et al., 1996):

$$T_{i,r} = \sum_{j=1}^{n} E_{j,r} \tag{4}$$

where j stands for different tree types of forest specie i; $E_{j,r}$ is the biomass of tree type j in province r.

$E_{j,r}$ was calculated based on the forest stock volume as following (Fang et al., 1996):

$$E_{j,r} = aV_{j,r} + b \tag{5}$$

Where $E_{j,r}$ is the biomass of tree type j in province r; $V_{j,r}$ indicates the forest stock volume (m$^{-3}$) of tree type j in province r; a and b were set as correlation coefficient.

"a" and "b" for different tree types were derived from previous studies (Fang et al., 1996; Tian et al., 2011; Lu et al., 2012; Li et al., 2014; Wang et al., 2014; Wen et al., 2014) (Table 2 in manuscript), $V_{j,r}$ was collected from the 8th Chinese National Forest Resource Inventory.

In our study, the correlation coefficient "a" and "b", the calculated method and the 8th Chinese National Forest Resource Inventory were all proved to be reliable (Fang et al., 1996; Lu et al., 2012). Therefore, the result of the biomass data in our study is justifiable. However, the correlation coefficient "a" and "b" in current studies have not shown the difference in the same tree species of each region, which may cause uncertainty in results. We will improve this in our future research.

Sorry for the unclear description in the original manuscript. We have made the revision correspondingly in the manuscript (Line 182-187).

Lines 188-189: What does "similar" mean here?

**Response:**

Sorry for the confusing sentence. "similar" in line 188-189 means researches which used emission factors (EFs) of cropland, forest, shrubland and grassland in China or foreign regions. We added these EFs in our calculation due to the shortage of EFs for some crop species burning in CEC and the shortage of EFs for pollutants emitted from forest, grassland and shrubland burning in China. To avoid misleading information, we have changed "similar research" to "previous researches" in the manuscript (Line 224).

Line 190: Is there a reference for this previous research?

**Response:**

Thanks for this comment. Emission factors of OC and EC for corn straw burning were derived from our experimental data which have not been published now (as shown in following Table 1). The corn straw is from Nanjing, which is one of the most important cities in CEC.

Meanwhile, we are so sorry for a mistake in Table 5. Table 5 in the original manuscript was a temporary table due to we tried to use our own EFs of OC and EC for forest fire burning in CEC. The OC (2.6 g/kg) and EC (0.11 g/kg) of coniferous forest were measured for pine wood and the OC (1.1 g/kg) and EC (0.31 g/kg) of broadleaf forest were measured for cotton wood (as shown in following Table 1). However, we found that the open wood burning in our experimental were not truly equal to forest fire. Therefore, we choose the emission factors of forest fire from Akagi et al. (2011) in our emission estimation. Emission factors and emission results for forest fire in our study were corresponding to Akagi et al. (2011), instead of our own data. We forget to update Table 5 in the original manuscript until we found this problem recently. We are so sorry for this mistake. We have made the revision in the manuscript (Table 5).

**Table 1.** The original experimental data for EFs calculating

| Types | fuel consumption/kg | dilution ratio | Particle size/μm | Sampling flow | The number of sampling film | OC (ug) | EC (ug) |
|---|---|---|---|---|---|---|---|
| Pine wood | 0.325 | 60 | 9.0~10.0 | 28.3 | XQ116 | 52.2496 | 1.5072 |
| | | 60 | 5.8~9.0 | | XQ117 | 31.6512 | 1.5072 |
| | | 60 | 4.7~5.8 | | XQ118 | 45.7184 | 1.5072 |
| | | 60 | 3.3~4.7 | | XQ119 | 51.2448 | 1.5072 |
| | | 60 | 2.1~3.3 | | XQ120 | 53.7568 | 1.5072 |
| | | 60 | 1.1~2.1 | | XQ121 | 87.92 | 1.5072 |
| | | 60 | 0.65~1.1 | | XQ122 | 396.896 | 2.0096 |
| | | 60 | 0.43~0.65 | | XQ123 | 632.0192 | 1.5072 |
| | | 60 | ≤0.43 | | XQ124 | 693.312 | 1.5072 |
| Cotton wood | 0.31 | 60 | 9.0~10.0 | 28.3 | XQ125 | 40.6944 | 1.5072 |
| | | 60 | 5.8~9.0 | | XQ126 | 19.0912 | 1.5072 |
| | | 60 | 4.7~5.8 | | XQ127 | 21.1008 | 1.5072 |
| | | 60 | 3.3~4.7 | | XQ128 | 33.6608 | 1.5072 |
| | | 60 | 2.1~3.3 | | XQ129 | 21.1008 | 1.5072 |

| | | | | | | | |
|---|---|---|---|---|---|---|---|
| | | 60 | 1.1~2.1 | | XQ130 | 46.2208 | 7.536 |
| | | 60 | 0.65~1.1 | | XQ131 | 37.1776 | 19.0912 |
| | | 60 | 0.43~0.65 | | XQ132 | 80.384 | 16.0768 |
| | | 60 | ≤0.43 | | XQ133 | 485.3184 | 118.064 |
| Corn straw | 0.31 | 60 | 9.0~10.0 | 28.3 | XQ143 | 67.824 | 1.5072 |
| | | 60 | 5.8~9.0 | | XQ144 | 40.192 | 1.5072 |
| | | 60 | 4.7~5.8 | | XQ145 | 36.1728 | 1.5072 |
| | | 60 | 3.3~4.7 | | XQ146 | 47.2256 | 1.5072 |
| | | 60 | 2.1~3.3 | | XQ147 | 36.1728 | 1.5072 |
| | | 60 | 1.1~2.1 | | XQ148 | 39.1872 | 1.5072 |
| | | 60 | 0.65~1.1 | | XQ149 | 67.3216 | 4.5216 |
| | | 60 | 0.43~0.65 | | XQ150 | 99.4752 | 1.5072 |
| | | 60 | ≤0.43 | | XQ151 | 601.8752 | 59.7856 |

Line 193:"opening"-"open"

**Response:**

It has been corrected (Line 228).

Line 202: Maybe the title should be "Other factors influencing OBB emission"?

**Response:**

Thanks for your suggestion. It has been corrected (Line 237).

Line 209:"20, 000"-"20,000"

**Response:**

It has been corrected (Line 244).

Lines 214-215: It should be made clear somewhere that $PM_{2.5}$ and OC/EC are not totally different things, i.e. PM will include some OC and EC.

**Response:**

Thank you very much for this query. We are really sorry for the mistake. It is really a clerical error. Emission of $PM_{2.5}$ was $4.13\times10^3$ Gg instead of $4.13\times10^2$ Gg. It has been corrected in the text (Line 251 ).

Line 217: Suggest removing "section".

**Response:**

It has been deleted (Line 251).

Line 240:"This is mostly": Which? For example, what are the "suitable weather conditions" in this case?

**Response:**

Thank you very much for this query. "This" means the relative high contributions of forest and shrubland fire burning emissions in southern provinces. Corresponding correction has been done in the text as following (Line 275-277).

*"The relative high emission contributions of forest and shrubland fire burning in the southern provinces can be explained by the large forest and shrubland coverage, frequent human forestry activities, low precipitation and dry weather in spring and winter (Cao et al., 2015), which may easily lead to forest and shurbland fires."*

Figure 6:"opening"-"open"

**Response:**

It has been corrected.

Lines 255-259: There is confusion between trends and short-term (interannual) variations here. Can the authors perhaps provide the overall trend for each species for the whole period, and whether it is statistically significant?

**Response:**

Thanks for this query. The variation tendency of different pollutants and different biomass types were shown in following Figure 1. For different pollutant species, we found that the emissions have similar increasing tendency ($R^2$ higher than 0.32, P<0.05). For different biomass types, cropland fire burning emission have a significant increasing tendency ($R^2$=0.87, P<0.01), while no significantly variation trend was observed for forest, shrubland and grassland fire burning emissions in research period (P>0.05).

On the process of data analysis, we think discussing the interannual variation is more meaningful than listing the changing tendency during the whole research period. It could be found that the increase of crop residue burning dominated the significant growth of OBB emission from 2003 to 2008. Then with the adoption of strict control policies (Table S1 in Supplementary file), the growth of crops residue burning emission gradually slow down. Meanwhile, the forest, shrubland and grassland fire burning were mainly affected by weather conditions and human activities as discussed above. Their emissions were difficult to predict and control, and exhibited random yearly variation of pollutant emission. Therefore, we initially discussed the mulit-year variation during 2003-2015 instead of the overall trend for the whole period in this study.

We adopted the suggestion and added these discussions in the text (Line 296-301). And the following Figure 1 was also added in the supplementary file (Figure S3).

[Figure]

**Figure 1.** The changing tendency of different pollutant emissions (a) and $PM_{2.5}$ emission from different biomass types (b) during 2003 to 2015.

Line 265: Presumably it is just coincidental that they are odd and even years. Therefore maybe not worth the emphasis?

**Response:**

Thanks for this query. Sorry for this unclear description. As discussed above, the forestry fire burning was affected by environmental conditions and human activities with environmental factors having a larger impact (Chen et al., 2013), so their emissions are random. Initially, it is just a general description (Line 312-313). We have corrected the sentences as *"with higher emission in 2011, 2013 and 2015 and lower emission in 2012 and 2014."*

Line 280: Increased or decreased?

**Response:**

Sorry for this mistake, it has been corrected as *"...increasing trend from 2003 to 2015, by about 21%-29%."* (Line 334).

Line 288:"scattering"-"scattered"

**Response:**

It has been corrected (Line 343).

Line 315:"totally occupied. . .": Please improve phrasing.

**Response:**

It has been corrected as following:

*"However, the emissions of $PM_{2.5}$ from forest, shrubland and grassland burning achieved peak values from December to March, being 67% of that in 2003-2015."* (Line 372-373)

Line 318:"based on the correlation of emissions in each month"-"based on the correlation between their monthly emissions"

**Response:**

It has been corrected (Line 378).

Line 328:"occurred"-"occurring"

**Response:**

It has been corrected (Line 389).

Line 345:"were found"-"featured"

**Response:**

It has been corrected (Line 407).

Lines 348-349:"are uniformed"-"remain similar"

**Response:**

It has been corrected (Line 409-410).

Line 378: May need to improve terminology a bit. For example, why is "local burning habits" not part of "anthropogenic activities"? What is the distinction?

**Response:**

Thanks for this query. To our knowledge, "burning habit" lead to anthropogenic activity, which is not anthropogenic activity itself. Sweep graves in tomb-sweeping day and celebration in Spring Festival are anthropogenic activities, which caused by social customs. In order to avoid unclear clarification, we change "some anthropogenic activities" to "social customs" through the full text in the revised manuscript.

Line 380: "People sweep their graves": The authors may want to change the wording as "their" might be a bit misleading here...

**Response:**

Thank you very much for your comment. We are truly sorry for the misunderstanding. We have made the revision in revised manuscript (Line 441).

*"People sweep graves and burn sacrifices by ignited straw..."*

Line 382:"file"-"fire"

**Response:**

It has been corrected (Line 443).

Line 406:"where"-"which"

**Response:**

It has been corrected (Line 468).

Line 416:"have"-"having"

**Response:**

It has been corrected (Line 478).

Line 420: Maybe "close" is a bit of an overstatement? They should not be expected to be very close anyway as there seem to have been several improvements in the current study.

**Response:**

Thank you very much for this comment. We agree with you that they should not be expected to be very close anyway as there are several improvements in this study. Actually, we use "close" to describe the objective results instead of emphasizing the similarity of the results. The EFs employed in Wang et al. (2008) were constant values for different biomass species burning. The interaction between some undervalued EFs variables and overvalued EFs variables in Wang et al. (2008) may lead to this "close" result. As shown in following Table 2, the uncertainty range of different pollutant species were much smaller than those in Wang et al. (2008), which verified that our results were improved. We are so sorry for the confusing description. We have made the revision in the manuscript (Line 482-483).

Table 2. Uncertainty ranges of different pollutants in emission estimates

| Pollutant | Uncertainty ranges | |
|:---:|:---:|:---:|
| | Our study | Wang et al., 2008 |
| OC | ±30% | ±148% |
| EC | ±48% | ±132% |
| $CH_4$ | ±20% | ±77% |
| $NO_X$ | ±20% | ±80% |
| NMVOC | ±45% | ±71% |
| $SO_2$ | ±45% | ±108% |
| $NH_3$ | ±35% | ±137% |
| CO | ±18% | ±86% |
| $CO_2$ | ±3% | ±60% |
| $PM_{2.5}$ | ±36% | ±142% |

Line 421-422:"The differences were mainly caused. . .": How do the authors know that these are the causes of differences? And which of the factors may be more responsible?

**Response:**

Thank you very much for your comment. By comparing with different methods and parameters (e.g. biomass loading data, EFs) involved in the processing of emission estimation in previous studies, we find the reasons for the differences in our research. In previous studies, the differences were resulted from different methods and parameters adopted. Compared with previous studies, the combination of multiple satellite products with local EFs data and updated local biomass loading data are likely to improve the estimation of pollutant emission from OBB effectively in this study. Please see the detailed response to the "General Response" above. In order to give a clearer description, we have made revision in the manuscript (Line 485-489).

Line 431: Are lower EFs used here more justifiable?
**Response:**

Thanks for this query. We are sorry for the unclear description. The justifiable of EFs used in our study is not because of the lower values, but some of EFs were collected from previous research carried out in CEC (Tang et al, 2014), which were more accurate and suitable in CEC. It has been corrected in the text (Line 498-499).

Lines 436-439: This can be combined into one sentence to avoid repetition.

**Response:**

We have made corresponding revision in the manuscript (Line 506-509).

Line 438:"localized in CEC"-"specific to CEC"

**Response:**

The sentence has been deleted and combined according to the above suggestion.

Line 439:"can improve"-"are likely to have improved"

**Response:**

It has been corrected (Line 508).

Line 446: Do 0.03 and 0.85 refer to relative uncertainties?

**Response:**

Sorry for the unclear description. The uncertainty for the emission factors was cited from references in the supplementary Table S5, mainly ranged from 0.03-0.85 for each pollutant.

Line 446:"At last"-"Finally"

**Response:**

It has been deleted.

Line 474:"opening"-"open"

**Response:**

It has been corrected (Line 558).

Line 487:"of the next year" is not needed.

**Response:**

It has been deleted (Line 572).

Line 488:"impacted on the emission": This implies causality but the analysis has been based on correlations, which can only be suggestive of possible associations but not conclusive for cause-effect relationships.

**Response:**

Thank you very much for this suggestion. We quite agree with your opinion that possible associations does not means conclusive for cause-effect relationships. The sentences have been corrected as following (Line 572-574).

[revised manuscript text omitted]

---

## Author Response (AR2)

**Authors' Responses to Reviewer Comments**

Manuscript: Estimating the open biomass burning emissions in Central and Eastern China from 2003 to 2015 based on satellite observation (Ref. No.: acp-2018-282)

We are very grateful for your careful and insightful comments, which contributed greatly to improve this manuscript. We carefully answered them point-by-point as below and corrected the corresponding parts in the manuscript (in red color).

Please make sure that all the references are correct. For example, for Wiedinmyer et al. (2011) please make sure you cite the final Geosci. Model. Dev. paper (not the Geosci. Model. Dev. Discuss. paper).

Response:

Thanks for this suggestion. We have corrected it in the manuscript (in Line 794). In addition, we have checked all the references carefully during the revision of our paper.

We have carefully taken the editor's suggestion into consideration. Please see the following point-by-point responses.

Line 28-29: Change "From 2003 to 2015, the emissions from forest, shrubland and grassland fire burning had little interannual variability"

Response:

We really appreciate for this comment. We have made corresponding revision in the manuscript.

*"From 2003 to 2015, the emissions from forest, shrubland and grassland fire burning held annual fluctuation whereas the emissions from crop straw burning steadily increased."* (Line 28-29)

Line 40: What is the meaning of "whole"? Do you mean total?

Response:

Sorry for the confusing sentence. the whole pollutants emission means the total pollutant emission in research period. We have changed "the whole pollutants emission" to "the total pollutant emission" in the manuscript (Line 39).

Line 46: Change "influence" to "influencing"

Response:

It has been corrected (Line 45).

Line 67: "hundreds of million of open biomass burned": the meaning here is not clear.

Response:

Sorry for the confusing sentence. We have made revision in the manuscript.

[revised manuscript text omitted]